# First-line Nivolumab plus FOLFOXIRI/Bevacizumab in advanced RAS/BRAF-mutated colorectal cancer: efficacy, safety and biomarker discovery from the phase II NIVACOR trial

Immunotherapy achieved remarkable results in patients with deficient mismatch repair (dMMR)/microsatellite instable (MSI) metastatic colorectal carcinoma (mCRC). However, its efficacy in proficient MMR (pMMR)/microsatellite stable (MSS) mCRC remains limited. In the phase II NIVACOR trial, we evaluated the activity and safety of FOLFOXIRI/bevacizumab plus nivolumab as first-line therapy in patients with RAS/BRAF-mutated mCRC (NCT04072198). The primary endpoint of the trial was the Objective Response Rate (ORR) whereas secondary endpoints were safety profile, overall survival (OS), progression free survival (PFS), duration of response (DoR) and quality of life. The primary endpoint was met. Among the 73 enrolled patients, 76.7% achieved an objective response (95% CI, 65.4 to 85.8%), while the disease control rate was 97.3% (95% CI, 90.5 to 99.7%). The median progression-free survival (mPFS) was 10.1 months (95% CI, 9.0 to 14.3 months), and the median overall survival (mOS) was not reached. Treatment-related adverse events of grade 3 or higher occurred in 48 patients out 73 enrolled patients (65.8%). Comprehensive genomic profiling and RNA sequencing analysis revealed genomic and transcriptomic profiles associated with treatment response in pMMR/MSS patients. Alterations in pathways such as PI3K/AKT, chemokine signaling and DNA repair showed correlation with treatment activity. These findings highlight the potential synergy between immune checkpoint inhibitors and cytotoxic chemotherapy in selected patients with pMMR/MSS mCRC.

Colorectal cancer (CRC) accounts for 10% of global cancer incidence, and about 9.4% of cancer-related deaths[1]. ~15–30% of patients at diagnosis present synchronous metastases while 20–50% will develop metachronous metastases[2].

The standard treatment for metastatic CRC (mCRC) is cytotoxic chemotherapy, comprising fluoropyrimidines, oxaliplatin, and irinotecan[3]. The phase III TRIBE trial indicated that triplet chemotherapy with FOLFOXIRI combined with bevacizumab was more effective than a doublet chemotherapy regimen with bevacizumab in the first-line setting, significantly improving progression-free survival (PFS) (12.1 months vs 9.7 months; HR 0.75; 95% confidence interval [CI], 0.62 to 0.90; p = 0.003), overall survival (OS) (29.8 months vs

✉ e-mail: nicola.normanno@irst.emr.it; nicnorm@yahoo.com

25.8 months; HR 0.80; 95% CI 0.65–0.98; $p = 0.03$), and objective response rate (ORR) (65% vs 53%; $p = 0.006$)[4]. However, given the increased toxicity associated with triplet chemotherapy, careful patient selection is essential for this approach.

Microsatellite instability (MSI), caused by genetic or epigenetic alterations leading to a deficient mismatch repair (dMMR) system, accounts for about 15% of all CRCs and only 4-8% of mCRC[5–7]. Several trials demonstrated the efficacy of immune checkpoint inhibitors in patients with dMMR/MSI mCRC. In particular, the KEYNOTE-177 phase III trial showed that in chemo-naïve patients with dMMR/MSI mCRC, single-agent pembrolizumab versus standard chemotherapy significantly improved PFS (16.5 vs 8.2 months; HR 0.59; $p = 0.0002$) and ORR up to 43.8%. Overlapping results arise from the GARNET trial of single-agent dostarlimab in dMMR/MSI mCRC, achieving an ORR of 43.5% (95% CI 34.3–53.0), and a median duration of response (DoR) not reached[8,9]. Interestingly, the combination of nivolumab plus low-dose ipilimumab showed a higher response rate as compared with single-agent checkpoint inhibitor (ORR = 69%; 95% CI 53–82 and Disease Control Rate, DCR = 84%; 95% CI 70.5–93.5) in patients with dMMR/MSI mCRC, with 13% of complete responses[10].

Although immunotherapy has shown remarkable efficacy in dMMR/MSI mCRC, its success remains limited in patients with proficient mismatch repair/microsatellite stable (pMMR/MSS) mCRC[11]. In order to improve the efficacy of immunotherapy in this setting, several clinical trials explored the efficacy of combinations of immune checkpoint inhibitors, chemotherapy and/or anti-angiogenic drugs. These latter agents may favor the trafficking of T cells by normalizing the tumor vasculature and have direct effects on immune response by reducing the frequency of myeloid-derived suppressor cells (MDSC), decreasing suppressive cytokines and tumor-infiltrating Tregs and MDSCs, and increasing dendritic cell maturation and function[12]. In particular, the randomized, open-label phase II AtezoTRIBE study of first-line FOLFOXIRI plus bevacizumab with or without atezolizumab demonstrated a statistically significant improvement in terms of PFS in the atezolizumab group compared with the control group (13.1 months vs 11.5 months, HR 0.69; 80% CI 0.56-0.85; $p = 0.012$; adjusted HR 0.70; 80% CI 0.57-0.87; log-rank test $p = 0.018$), without advantage in ORR by the addition of atezolizumab to FOLFOXIRI plus bevacizumab in an unselected population of not previously treated patients with mCRC[13]. Interestingly, in the pMMR subgroup, the median PFS was 13.0 months vs 11.5 months (HR 0.80; 80% CI 0.65-0.99; $p = 0.088$)[14].

This manuscript presents results from the phase II NIVACOR trial, which evaluated the activity and safety of FOLFOXIRI/bevacizumab combined with nivolumab as first-line therapy in patients with RAS/BRAF-mutated mCRC, a population with poor prognosis. We also show the results of comprehensive genomic profiling (CGP) and RNA-sequencing (RNAseq) analyses performed to identify possible biomarkers of activity of this combination in pMMR/MSS tumors.

## Results

### Clinical activity in the overall and subgroup populations

Between October 19, 2019, and March 4, 2021, 73 patients with advanced, unresectable, treatment-naïve RAS/BRAF-mutated mCRC were enrolled and received study treatment in 11 sites in Italy. Baseline demographic and disease characteristics of the whole cohort are listed in Table 1 and Supplementary Data.

As per protocol analysis, 65 RAS mutations and 11 BRAF V600E mutations were detected in 73 patients by local pathology assessment. Concomitant RAS and BRAF V600E mutations were reported in 3 patients. Ten patients (13.7%) carried a dMMR/MSI CRC, while 63 (86.3%) were pMMR/MSS. The median number of cycles received was 13 (range 1–46); the median treatment duration was 7.5 months (0-24 months), with 4.2 months (range 0–9.5 months) and 8.7 months (range 4.4–24 months) for induction and maintenance treatment,

**Table 1 | Baseline demographic and disease characteristics in the Overall Population**

| Characteristics | N = 73; N (%) |
|---|---|
| **Median age, years (range)** | 60 (51-65) |
| **Gender** | |
| Male | 37 (50.7) |
| Female | 36 (49.3) |
| **Geographic region** | |
| Caucasian | 70 (95.9) |
| Asia | 2 (2.7) |
| North Africa | 1 (1.4) |
| **ECOG performance status** | |
| 0 | 59 (80.8) |
| 1 | 14 (19.2) |
| **Primary tumor site** | |
| Right | 41 (56.2) |
| Left | 32 (43.8) |
| **Number site metastases** | |
| 1 | 40 (54.8) |
| 2 | 15 (20.5) |
| ≥3 | 18 (24.7) |
| **Site metastases** | |
| Liver | 41 (56.2) |
| Peritoneum | 24 (32.9) |
| Lymph nodes | 26 (35.6) |
| Lung | 13 (17.8) |
| **Previous adjuvant chemotherapy** | 13 (17.8) |
| **Previous radiotherapy** | 2 (2.7) |

*ECOG* Eastern Cooperative Oncology Group.

respectively. The median follow-up was 24 months (95% CI 23.9–24), and 64 patients discontinued experimental treatment. The main reason for discontinuation was disease progression (39 patients; 61.0%).

The primary endpoint was met. Among the overall population, 56 patients (76.7%) achieved ORR events (95% CI, 65.4% to 85.8%), including 7 complete responses (CR; 9.6%) and 49 partial responses (PR; 67.1%) (Fig. 1a; and Table 2).

Fifteen out of 73 patients (20.6%) had stable disease (SD), while two patients were not evaluable for the primary endpoint (Table 2). The DCR was 97.3% (95% CI 90.5 to 99.7%). The median time to response was 2.1 months (IQR, 1.8-3.7 months), and the median DoR was 7.4 months (95% CI 6.0 to 13.6 months) (Fig. 1b). Changes from baseline in tumor size showed a general reduction in tumor burden over time, also in some patients with SD (Fig. 1c). The subsequent surgery of the primary tumor, metastases, or both, was performed on 6 (8.2%), 9 (12.3%), and 5 (6.9%) patients, respectively.

At the end of the study, 54 (74%) events of PFS were observed, of which 48 were PD. The median PFS (mPFS) was 10.1 months (95% CI, 9.0 to 14.3 months) (Table 2; and Fig. 1d). Deaths occurred in 28/73 patients (38.4%). The median OS was not reached (Fig. 1e).

Subgroup analysis revealed that the RAS mutant and the BRAF mutant subgroups had an ORR of 75.8% and 81.8% (Table 2 and Fig. 1a) and a median DoR of 7.2 months and not reached, respectively (Table 2; and Supplementary Fig. 1a). The mPFS was 9.7 months and not reached for patients with RAS and BRAF mutations, respectively (Table 2; and Supplementary Fig. 1b). The median OS was reached in neither subgroup (Table 2; Supplementary Fig. 1c). Information on the specific KRAS mutation of enrolled patients was available for 47

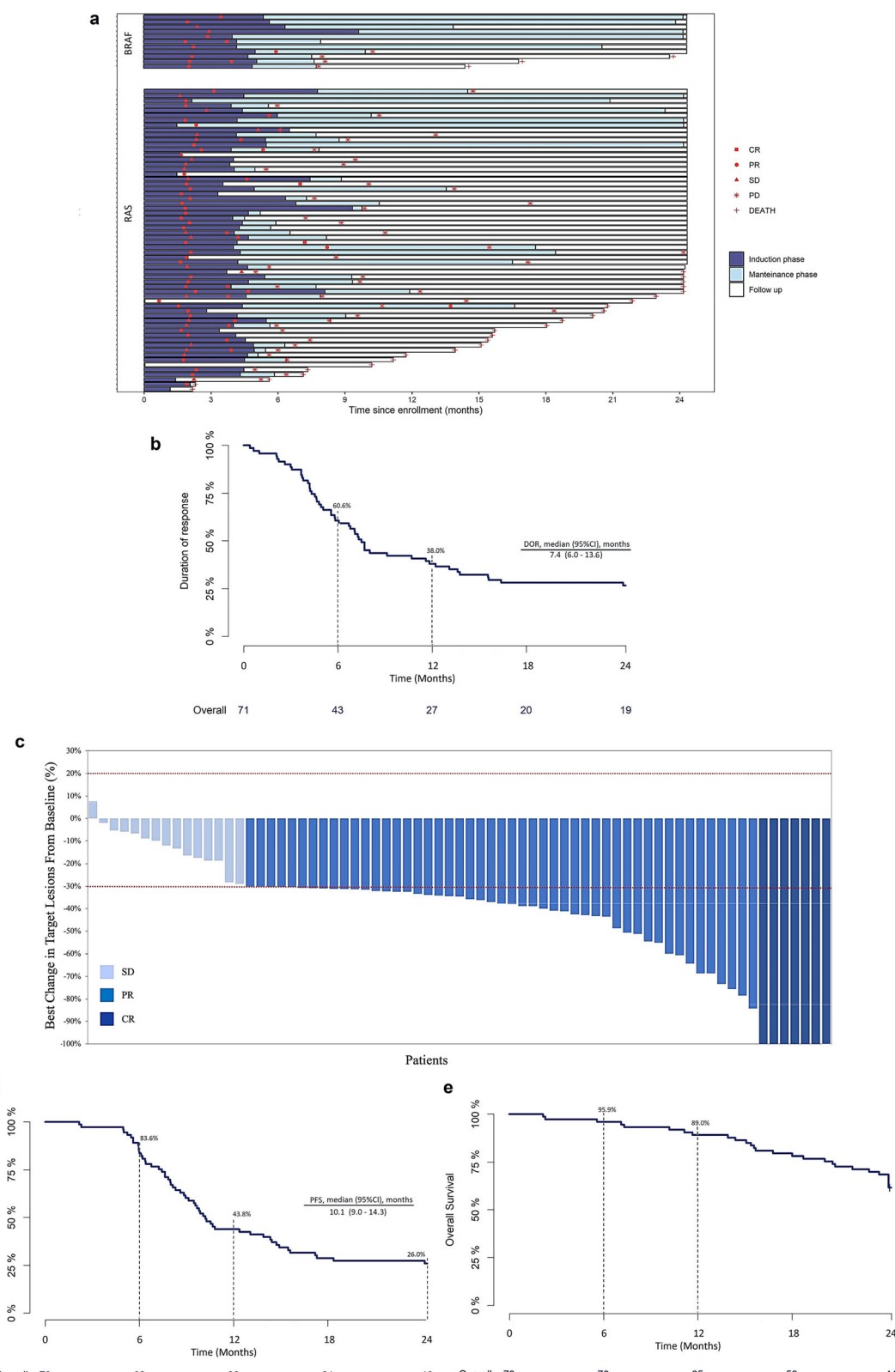

patients, based on translational analyses performed in the study. Of these, 34 patients had mutations in *KRAS* codon 12 (G12X) and 13 in other codons (non-G12X). The mPFS of G12X and non-G12X patients was 9.55 and 12.9 months, respectively ($p = 0.6$). (Supplementary Fig. 1d).

In the 63 pMMR/MSS and 10 dMMR/MSI patients' cohorts, the ORR was 77.8% and 70.0%, respectively (Table 2; Supplementary Fig. 2a). The DCR was 96.8% and 100% (Table 2, Supplementary Fig. 2a) and the DoR 7.0 months and not reached, respectively (Table 2; Supplementary Fig. 2b). The mPFS of the pMMR/MSS subgroup was 9.5 months and not reached in dMMR/MSI patients, respectively (Table 2 and Supplementary Fig. 2c), and the median OS was reached in neither subgroup (Table 2 and Supplementary Fig. 2d).

**Fig. 1 | Tumor response and survival analysis in the overall population of the NIVACOR study. a** Plot of time to response and death in individual RAS and BRAF mutated mCRC patients ($n = 73$). Blue bars indicate treated patients in the induction phase with FOLFOXIRI/bevacizumab plus nivolumab; light blue bars indicate treated patients in the maintenance phase with bevacizumab plus nivolumab. **b** Kaplan-Meier curve of the duration of response (DoR) for the overall population treated with FOL-FOXIRI/bevacizumab plus nivolumab. **c** Waterfall plot of best response with maximum percent decrease from baseline in the sum of diameters of target tumors based on independent central review assessment per RECIST version 1.1 criteria. The dotted lines denote a 30% decrease and a 20% increase in tumor size cutoffs for partial response and progressive disease, respectively. **d, e** Kaplan-Meier curves of progression-free survival (**d**) and overall survival (**e**) in the whole cohort of patients enrolled in the NIVACOR trial. 95% CI values were calculated using the Clopper Pearson method. CR complete response; PR partial response; SD stable disease; PD progression disease.

**Table 2 | Response Rate per Independent Central Review and Survival Outcomes in the Overall Population and by Patient Subgroup**

|  | Overall Population N = 73; N (%) | RAS mut subgroup N = 62; N (%) | BRAF mut subgroup N = 11; N (%) | MSS subgroup N = 63; N (%) | MSI subgroup N = 10; N (%) |
|---|---|---|---|---|---|
| ORR; 95% CI | 56 (76.7); 65.4 to 85.8 | 47 (75.8); 65.3 to 85.8 | 9 (81.8); 48.2 to 97.7 | 49 (77.8); 65.5 to 87.3 | 7 (70.0); 34.8 to 93.3 |
| CR | 7 (9.6) | 5 (8.1) | 2 (18.2) | 6 (9.5) | 1 (10) |
| PR | 49 (67.1) | 42 (67.7) | 7 (63.6) | 43 (68.2) | 6 (60) |
| SD | 15 (20.6) | 13 (21.0) | 2 (18.2) | 12 (19.1) | 3 (30) |
| No assessment * | 2 (2.7) | 2 (3.2) | 0 (0) | 2 (3.2) | 0 (0) |
| DCR (CR + PR + SD); 95% CI | 71 (97.3); 90.5 to 99.7 | 60 (96.8); 88.8 to 99.6 | 11 (100); 71.5 to 100 | 61 (96.8); 89.0 to 99.6 | 10 (100); 69.2 to 100 |
| Median DoR, months; 95% CI | 7.4; 6.0 – 13.6 | 7.2; 5.5 – 11.4 | NR; 6.0-NE | 7.0; 5.0- 9.0 | NR |
| Median PFS, months; 95% CI | 10.1; 9.0 to 14.3 | 9.7; 8.7 to 13.7 | NR; 10.1 to NE | 9.5; 8.2 to 12.2 | NR |
| Median OS, months | NR | NR | NR | NR | NR |

*mut* mutated, *MSS* microsatellite stable, *MSI* microsatellite instable, *ORR* Overall Response Rate, *CR* complete response, *PR* partial response, *SD* stable disease, *DCR* disease control rate, *DoR* duration of response, *NR* not reached, *NE* not estimable, *PFS* progression-free survival, *OS* overall survival.

* Includes patients who did not perform the first assessment.

## Safety

A total of 449 treatment-related adverse events (TrAEs) of any grade were observed across the three study drugs (Supplementary Table 1). Sixty-three out of 73 patients (86.3%) experienced at least one TrAEs of any grade, and 48 patients (65.8%) experienced at least one grade 3-5 TrAE. Non-treatment-related AEs occurred in 48 of 73 patients (65.8%), indicating that TrAEs were more frequent than non-treatment-related ones. Any grade TrAEs in the overall population were reported in Supplementary Table 1.

Diarrhea (78.0%), fatigue (61.6%), neutropenia (58.9%), neurotoxicity (53.4%) and nausea (49.3%) were the most common any-grade TrAEs. Among these, diarrhea was mostly related to chemotherapy and nivolumab (45.2% and 32.8%, respectively), neutropenia mainly to chemotherapy (41.0%), and hypertension (15.0%) to bevacizumab.

Neutropenia (39.7%), diarrhea (23.2%), hypertension (8.2%), and fatigue (6.8%) were the most frequent grade 3-5 TrAEs reported (Supplementary Table 1). One patient (1.4%) had an ileo-urethral fistula (grade 5) and concurrent diarrhea related to Clostridium difficile infection after the first cycle of experimental treatment, leading to colostomy packaging and permanent treatment discontinuation. Furthermore, one (1.4%) patient developed infusion-related toxicity to oxaliplatin during the 1st and 2nd cycles of treatment, leading to permanent drug discontinuation.

Among all grade immune-related adverse events (irAEs), the most frequent were diarrhea (32.8%) and fatigue (26.0%), followed by neutropenia (17.8%), hypothyroidism (16.4%), and an increase in amylasemia and lipasemia values (15.0%). Considering grade 3-5 irAEs, neutropenia (13.6%) and diarrhea (6.8%) were more represented (Supplementary Table 1). One patient developed myasthenia gravis (grade 3), leading to permanent treatment discontinuation.

Among patients with pMMR/MSS tumors, 57 out of 63 (90.5%) experienced TrAEs of any grade. Among patients with dMMR/MSI tumors, the rate was 100% (10 out of 10). Considering irAEs, 46 of 63 (73.0%) MSS patients and 7 of 10 (70.0%) MSI patients experienced at least one such event.

## Comprehensive genomic profiling analysis

From the whole cohort of patients enrolled in the NIVACOR trial, FFPE material from the primary tumor of 68 cases (93.2%) was available for translational studies (Supplementary Table 2).

The sequencing success rate was 80.9% (55/68) for comprehensive genomic profiling (CGP) with the Oncomine Comprehensive (OCA) Plus Assay. The mPFS of patients successfully analyzed with CGP was 9.9 months.

CGP revealed the presence of at least one genomic alteration (GA) in all patients ($n = 55$). Overall, 982 GAs in 336 genes were detected (602 single nucleotide variants, 101 short insertion/deletions, 258 copy number variations, 21 loss of heterozygosity), with a mean of 17.85 GAs per patient (Supplementary Fig. 3a).

RAS/BRAF mutational status was confirmed for all cases. Forty-nine tumors were MSS and 6 MSI. The median Tumor Mutational Burden (TMB) was 6.71 mutations/Mb, with a range from 0.95 to 35.57 mutations/Mb. We categorized tumors in high versus low TMB, using the cut-off value of 10.4 mut/Mb, previously defined by comparing the OCA Plus Assay with the reference FoundationOne Test[15]. Among the 49 MSS patients, 9 had a high TMB ($\geq$10.4 mutations/Mb) and a mPFS of 18.1 months, whereas 40 had a low TMB and a mPFS of 9.35 months ($p = 0.083$) (Fig. 2a). The mPFS of the 6 MSI patients was not reached (Fig. 2a).

In order to identify GAs possibly associated with the outcome in MSS patients, we compared the genomic profile of the subgroup of patients with poor prognosis (12 patients with PFS below the 25th percentile - Q1 PFS) with the subgroup of patients with good prognosis (12 patients with PFS over the 75th percentile - Q4 PFS). We identified 51 genes altered in the Q1 PFS subgroup but not in the Q4 PFS subgroup, that represent a possible resistance (RES) signature (Supplementary Table 3). Genomic alterations in any of the genes of the RES signature were found also in 16/25 patients of the Q2 ($n = 8/13$ patients) and Q3 ($n = 8/12$ patients) PFS groups (Supplementary Fig. 3b). We next compared the PFS curves of

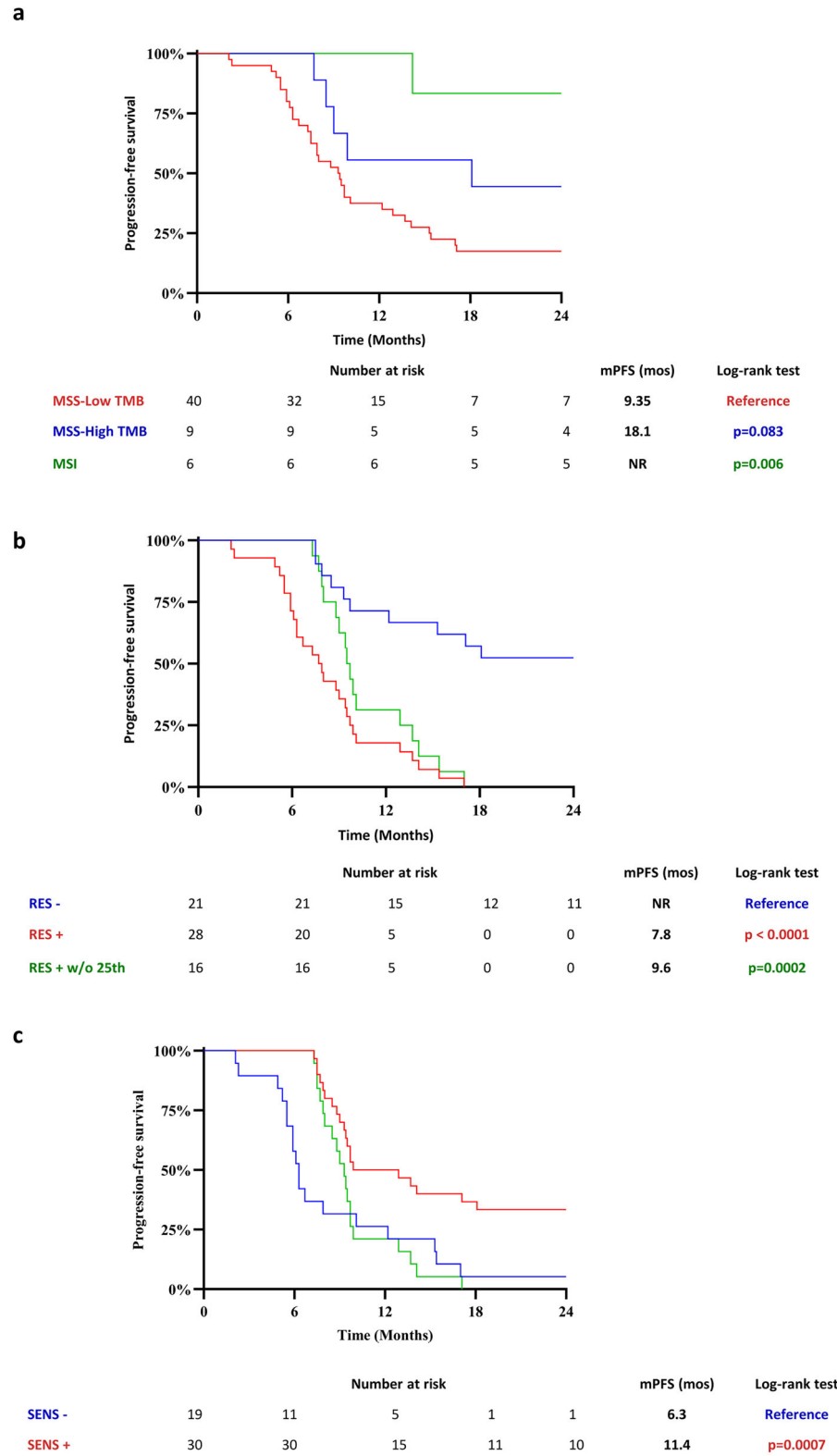

**Fig. 2 | Survival analysis of MSS/MSI patients according to TMB and RES/SENS signatures. a** Kaplan-Meier curves of PFS of patients analysed by Comprehensive Genomic Profiling according to TMB low (<10.4 mutations/Mb) and high (≥10.4 mutations/Mb) status for MSS patients (n = 49). Survival curve of PFS for MSI (n = 6) patients is also reported. **b** Kaplan-Meier curves of PFS stratified for MSS patients with GAs in at least one of the 51 genes of the RES signature (RES + ); for patients without GAs in any of them (RES—) and for RES+ patients in quartiles 2-3 of PFS (RES + without the 25th percentile; RES+w/o 25th). **c** Kaplan-Meier curves of PFS stratified for MSS patients with GAs in at least one of the 102 genes of the SENS signature (SENS + ); for patients without GAs in any of them (SENS—) and for SENS+ patients in quartiles 2-3 of PFS (SENS+ without the 75th percentile; SENS+w/o 75th). *P* values were calculated by the two-sided Log-rank (Mantel-Cox) test. mPFS median Progression-free Survival; RES resistance; SENS sensitivity; TMB Tumor Mutational Burden.

patients with GAs in at least one of the 51 genes of the RES signature (RES +) and patients without GAs in any of them (RES−). The 28 RES+ patients showed a significantly shorter PFS compared to the 21 RES− patients, with mPFS values of 7.8 months and not reached, respectively ($p < 0.0001$) (Fig. 2b).

When we excluded the poor prognosis subgroup (Q1 PFS) from the survival analysis, the 16 "RES+ without the 25th percentile" (RES+ w/o 25th) patients showed a worse PFS compared to the 21 RES− patients (mPFS values of 9.6 months and not reached, respectively; $p = 0.0002$) (Fig. 2b).

A gene set enrichment analysis performed using the Enrichr tool for genes in the RES signature showed an association with gene transcription and DNA repair pathways (Supplementary Table 4).

Conversely, we identified a potential sensitivity (SENS) signature possibly associated to a better outcome of MSS patients, selecting a group of 102 genes altered in the good prognosis subgroup (Q4 PFS) and not altered in the poor prognosis subgroup (Q1 PFS) (Supplementary Table 5). Genomic alterations in any of the genes of the SENS signature were carried also by 19/25 patients in the Q2 ($n = 12/13$ patients) and Q3 ($n = 7/12$ patients) PFS subgroups (Supplementary Fig. 3b). We then compared the PFS curves of patients with GAs in at least one of the 102 genes of the SENS signature (SENS + ) and patients without GAs in any of them (SENS−). The 30 SENS+ patients showed a significantly better PFS compared to the 19 SENS− patients, with mPFS values of 11.4 and 6.3 months, respectively ($p = 0.0007$) (Fig. 2c).When we excluded the good prognosis subgroup (Q4 PFS) from the survival analysis, the SENS signature showed a trend in separating the PFS curves of "SENS+ without the 75th percentile" (SENS+ w/o 75th) and SENS− patients (mPFS values of 9.30 and 6.30 months, respectively; $p = 0.560$) (Fig. 2c).

Enrichment analysis for the genes in the SENS signature revealed an association with different signal transduction pathways, including PIK3/AKT signaling (Supplementary Table 6).

## Transcriptomic analysis

RNAseq analysis was successfully performed for 48 out 68 patients (42 MSS and 6 MSI), with a success rate of the analysis of 70.6%, due to the low quality of RNA obtained from FFPE sections. The mPFS of the transcriptomic cohort was 9.7 months.

Among the 42 MSS patients, 11 were in the 25th PFS percentile subgroup and 9 in the 75th PFS percentile subgroup. We focused our attention on the differentially expressed genes (DEGs) between the subgroup of patients with better prognosis (Q4 PFS) and the subgroup of patients with the worse prognosis (Q1 PFS) in order to identify pathways potentially associated with the activity of the experimental treatment in MSS patients. In particular, we calculated the DEGs between these subgroups with fold change <-2.5 and >2.5 and a $p$ value = 0.0128. With these settings, we were able to discriminate the two subgroups in the PCA plot (Fig. 3a).

The association between the DEGs of the 25th and 75th percentile subgroups and the different clinical pathological features, such as localization of the tumor, presence of liver metastasis, and TMB values are shown in Fig. 3b.

The differentially expressed genes were 211, with 109 genes up-regulated and 102 down-regulated in the good prognosis 75th percentile PFS subgroup as compared with patients in the 25th percentile (Fig. 3c).

Gene Ontology analysis revealed for up-regulated genes an enrichment in pathways involved in the regulation of chemotaxis of neutrophils and of antigen receptor-mediated signaling pathway (Table 3) (adjusted $p$ value < 0.05).

Pathway enrichment analysis also showed an association with TGF-beta regulation of extracellular matrix, Binding of chemokines to chemokine receptors, G alpha (i) signaling events and PI3K cascade (Supplementary Table 7) (adjusted $p$ value ≤ 0.05).

For down-regulated genes, we found an association with DNA Replication-Dependent Chromatin Assembly in Gene Ontology analysis (Table 4) (adjusted $p$ value ≤ 0.05).

We also obtained protein-protein interaction (PPI) networks using the STRING database. For the DEGs upregulated in the 75th percentile, we found PPI networks with 95 nodes and 181 edges ($p = 1.0e-16$), showing an enrichment in genes involved in chemokines signaling and PIK3 pathways (Supplementary Fig. 4a). The PPI network obtained for down-regulated DEGs showed 85 nodes and 34 edges ($p = 2.88e-05$) and was enriched in genes involved in the CAF1 complex and protein-DNA complex assembly, and double strand-break repair system (Supplementary Fig. 4b).

As both genomic and transcriptomic analysis showed an enrichment in the PI3K pathway genes in the Q4 PFS subgroup, we performed a survival analysis on the whole cohort of patients analyzed by CGP ($n = 55$) and on the MSS subgroup ($n = 49$ patients), comparing patients with at least one GA in the PI3K/AKT pathway (PI3K/AKT MUT) and patients without GAs in the PI3K/AKT pathway (PI3K/AKT WT). The results showed in the whole cohort of 55 patients a mPFS of 13.7 months for the PI3K/AKT MUT patients versus 8.5 months for the PI3K/AKT WT patients ($p = 0.0431$ · Supplementary Fig. 5a). In the MSS subgroup, a mPFS of 9.9 months for the MUT versus 8.5 months for the WT patients was observed ($p = 0.1955$ – Supplementary Fig. 5b). Interestingly, the fraction of TMB-high patients was higher in patients with at least one GA in the PI3K/AKT pathway as compared to the WT patients both in the whole population (36.8% vs 5.5%; $p = 0.02$) and in the MSS cohort (25% vs 5.5%; $p = 0.14$). We next applied the consensus molecular subtype (CMS) classification, a validated signature based on gene expression profiling for CRC[16], to the cohort of 48 patients with available RNAseq data. Using the CMS classification, 41/48 patients (37 MSI and 4 MSS) were distributed among CMS1-4 subtypes and 7 remained unclassified (Supplementary Fig. 6a). The association between the CMS classification and the relative expression of signaling pathways was in agreement with previous findings[17], including the upregulation of TGF-Beta and EMT pathways in the CMS4 subgroup (Supplementary Fig. 6b).

Although the total number of cases was limited (n. 41 patients with known CMS status, 37 MSS and 4 MSI), we found a correlation between the CMS classification and PFS. In particular, patients within the CMS1 group showed a longer mPFS as compared with the other CMS classes (Supplementary Fig. 6c). However, in the cohort of 37 MSS patients with available CMS classification, no significant difference was found among the different CMS classes of CRC (Supplementary Fig. 6d).

We next combined the DNA and RNA data. DNA sequencing data were available for 36/37 patients with CMS classification. Detection of genomic alterations belonging to the RES signature identified subgroups with shorter PFS among the CMS2 ($p = 0.022$) and CMS4 ($p = 0.049$) and, at lesser extent, CMS1 ($p = 0.225$) and CMS3 ($p = 0.327$) cohorts (Fig. 4a). When the Q1 subgroup was excluded from the analysis, the same trend was observed within the different CMS classes, even though without statistical significance, due to the very low number of cases (Supplementary Fig. 7a). Although the low number of cases prevents any firm conclusion, these preliminary findings suggest that the RES signature can better stratify patients of the NIVACOR cohort as compared with the CMS classification. We performed a similar analyses for the SENS signature. The PFS of the SENS+ cases was generally longer as compared with the SENS− cases among the different CMS classes, although the difference is higher in the CMS1 group as compared the other CMS classes (Fig. 4b). A similar trend was observed when the Q4 PFS subgroup was excluded from the analysis (Supplementary Fig. 7b).

We also applied to our cohort of patients the Intrinsic CMS (iCMS) signature that better recapitulates tumor heterogeneity at single cell level[18]. Among the 48 patients with available RNAseq data, we classified 17 patients (16 MSS and 1 MSI) in the iCMS2 group and 17

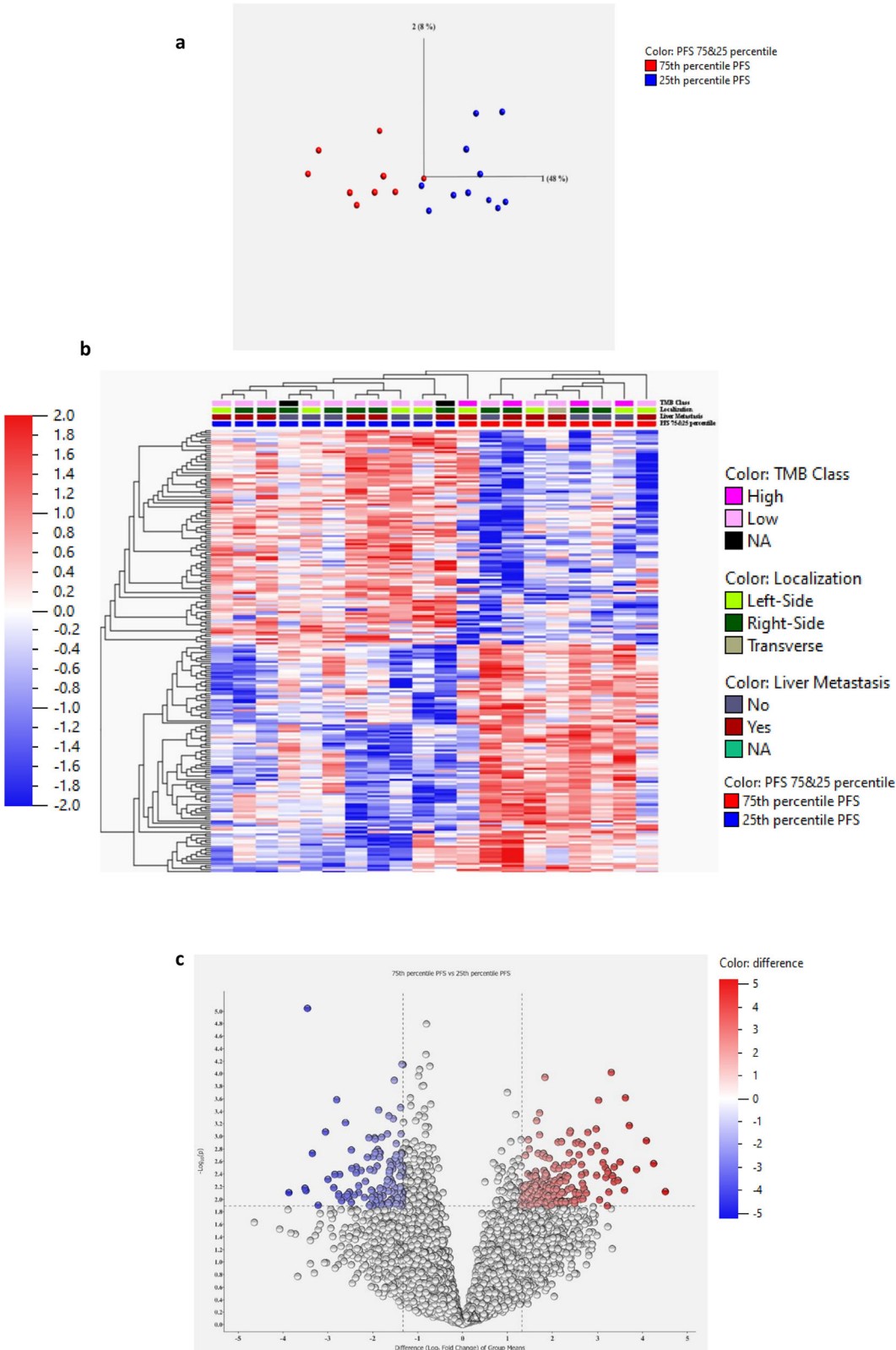

**Fig. 3 | RNAseq analysis of mCRC patients in the NIVACOR study.** RNAseq data on the 11 patients in the subgroup with the worse prognosis (25th percentile PFS) and on 9 patients in the better prognosis subgroup (75th percentile PFS) based on the PFS values were shown (**a**) Principal Component Analysis (PCA) plot of patients in the 25th percentile PFS (blue dots) and in 75th percentile PFS (red dots) subgroups. **b** Heatmap of differentially expressed genes between patients in the 25th percentile PFS ($n = 11$) and in 75th percentile PFS ($n = 9$). Information on TMB, localization of the tumor and presence of liver metastasis were included. **c** Volcano plot of differentially expressed genes between patients in the 75th and 25th PFS percentile subgroups. The differentially expressed genes were selected with fold change >2.5 or <−2.5 using t-test with $p$ value = 0.0128. PFS Progression-free Survival; TMB Tumor Mutational Burden.

**Table 3 | Gene Ontology enrichment analysis for up-regulated Differentially Expressed Genes**

| Index | Name | Adjusted p-value |
|---|---|---|
| 1 | Positive Regulation of Receptor-Mediated Endocytosis (GO:0048260) | 0.0001145 |
| 2 | Regulation of Receptor-Mediated Endocytosis (GO:0048259) | 0.001300 |
| 3 | Bone Development (GO:0060348) | 0.001300 |
| 4 | Positive Regulation of Endocytosis (GO:0045807) | 0.001706 |
| 5 | Negative Regulation of Cell-Matrix Adhesion (GO:0001953) | 0.003978 |
| 6 | Regulation of Keratinocyte Migration (GO:0051547) | 0.004069 |
| 7 | Negative Regulation of Focal Adhesion Assembly (GO:0051895) | 0.006652 |
| 8 | Negative Regulation of Cell-Substrate Junction Organization (GO:0150118) | 0.006652 |
| 9 | Negative Regulation of Protein Processing (GO:0010955) | 0.009025 |
| 10 | Positive Regulation of Neutrophil Chemotaxis (GO:0090023) | 0.01389 |
| 11 | Peptide Cross-Linking (GO:0018149) | 0.01485 |
| 12 | Positive Regulation of Granulocyte Chemotaxis (GO:0071624) | 0.01485 |
| 13 | Regulation of Focal Adhesion Assembly (GO:0051893) | 0.01485 |
| 14 | Positive Regulation of Intracellular Signal Transduction (GO:1902533) | 0.01579 |
| 15 | Negative Regulation of Cell Junction Assembly (GO:1901889) | 0.01580 |
| 16 | Chronic Inflammatory Response (GO:0002544) | 0.01580 |
| 17 | Response to Ketone (GO:1901654) | 0.01580 |
| 18 | Protein Localization to Cell Surface (GO:0034394) | 0.01580 |
| 19 | Positive Regulation of Neutrophil Migration (GO:1902624) | 0.01694 |
| 20 | Positive Regulation of Protein Phosphorylation (GO:0001934) | 0.01698 |
| 21 | Regulation of Neutrophil Chemotaxis (GO:0090022) | 0.01726 |
| 22 | Negative Regulation of Dendritic Cell Apoptotic Process (GO:2000669) | 0.01835 |
| 23 | Negative Regulation of Plasminogen Activation (GO:0010757) | 0.01835 |
| 24 | Regulation of Chemotaxis (GO:0050920) | 0.02321 |
| 25 | Regulation of Antigen Receptor-Mediated Signaling Pathway (GO:0050854) | 0.02356 |

Adjusted *p* values were calculated using the Benjamini-Hochberg method for correction for multiple hypotheses testing.

**Table 4 | Gene Ontology enrichment analysis for down-regulated Differentially Expressed Genes**

| Index | Name | Adjusted *p* value |
|---|---|---|
| 1 | DNA Replication-Dependent Chromatin Assembly (GO:0006335) | 0.0005407 |

Adjusted *p* values were calculated using the Benjamini-Hochberg method for correction for multiple hypotheses testing.

patients (15 MSS and 2 MSI) in the iCMS3 group. Fourteen patients were unclassified (Supplementary Fig. 8a). No significant differences in term of PFS were observed between the iCMS2 and iCMS3 groups in the whole cohort of 34 patients with iCMS classification (31 MSS and 3 MSI) (Supplementary Fig. 8b). The difference in PFS was not significant also when the cohort of 31 MSS patients was analyzed (Supplementary Fig. 8c). Finally, we combined the iCMS signature with CGP data in the MSS subgroup. Sequencing of DNA was successful for 30/31MSS patients with the iCMS classification. Detection of genomic alterations in the RES signature identified patients with a significantly low PFS in the iCMS3 group, as compared with RES— patients (6.0 months vs 21.05 months respectively, $p = 0.0007$) (Fig. 5a). No difference was observed in PFS of patients stratified for RES and iCMS2 signatures (9.25 months for patients RES+ versus 9.5 months for patients RES— in the iCMS2 cohort, $p = 0.235$) (Fig. 5a). When patients in the Q1 group were excluded from the analysis, a trend in lower PFS in the iCMS3 RES+ subgroup was observed (Supplementary Fig. 9a). For the SENS signature, the PFS of the SENS+ cases was higher as compared with the SENS— cases (18.1 months versus 6.1 months, respectively, $p = 0.068$) in the iCMS3 class (Fig. 5b). The difference was less marked for patients stratified for the SENS and the iCMS2 signatures (9.6 months for patients SENS+ versus

7.1 months for patients SENS—, $p = 0.454$) (Fig. 5b). When the Q4 cases were excluded from the analysis, a similar trend in PFS was observed although the difference was not statistically significant probably due to the low number of cases (Supplementary Fig. 9b). Our results suggest that both RES and SENS signatures better stratified patients in the NIVACOR study than the iCMS classification, although this observation should be confirmed in a larger number of patients.

## Discussion

The benefit of immune checkpoint inhibitors as monotherapy is primarily restricted to patients with dMMR/MSI mCRC. However, some dMMR/MSI patients do not respond to single-agent immunotherapy, suggesting the existence of mechanisms of intrinsic resistance to immune checkpoint inhibitors (ICIs) that might be shared with MSS tumors[19]. It has been hypothesized that adding chemotherapy or other agents to PD-1/PD-L1 blockade could overcome such resistance to ICIs[19]. Interestingly, a meta-analysis of studies of ICIs in mCRC suggested that anti-PD-1/PD-L1 agents combined with antiangiogenic drugs, targeted agents, and/or chemotherapy might be effective in patients with MSS mCRC[20]. In this respect, the results of the NIVACOR study show that adding nivolumab to FOLFOXIRI plus bevacizumab may improve the response rate in first-line treatment of patients with RAS/BRAF mutated mCRC, regardless of MMR status. Indeed, in the pMMR/MSS and dMMR/MSI patients' cohorts, the ORR was 77.8% and 70.0%, with a DCR of 96.8% and 100%, respectively. Interestingly, in dMMR/MSI patients treated with pembrolizumab (Keynote-177) or nivolumab plus ipilimumab (CheckMate-142), the best response was disease progression in 29% and 12% of the cases, respectively[9,10]. These data suggest that the dMMR/MSI population is heterogeneous and that some patients may benefit from more aggressive treatments, including chemotherapy[19]. The results of the NIVACOR trial are particularly

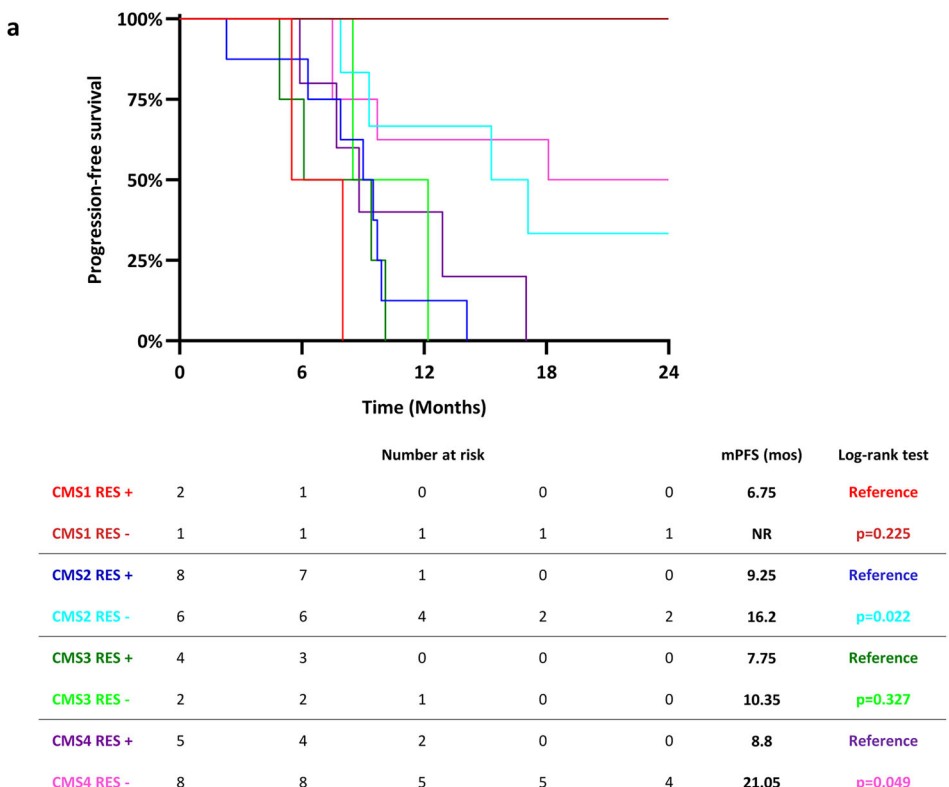

| | Number at risk | | | | | mPFS (mos) | Log-rank test |
|---|---|---|---|---|---|---|---|
| CMS1 RES + | 2 | 1 | 0 | 0 | 0 | 6.75 | Reference |
| CMS1 RES - | 1 | 1 | 1 | 1 | 1 | NR | p=0.225 |
| CMS2 RES + | 8 | 7 | 1 | 0 | 0 | 9.25 | Reference |
| CMS2 RES - | 6 | 6 | 4 | 2 | 2 | 16.2 | p=0.022 |
| CMS3 RES + | 4 | 3 | 0 | 0 | 0 | 7.75 | Reference |
| CMS3 RES - | 2 | 2 | 1 | 0 | 0 | 10.35 | p=0.327 |
| CMS4 RES + | 5 | 4 | 2 | 0 | 0 | 8.8 | Reference |
| CMS4 RES - | 8 | 8 | 5 | 5 | 4 | 21.05 | p=0.049 |

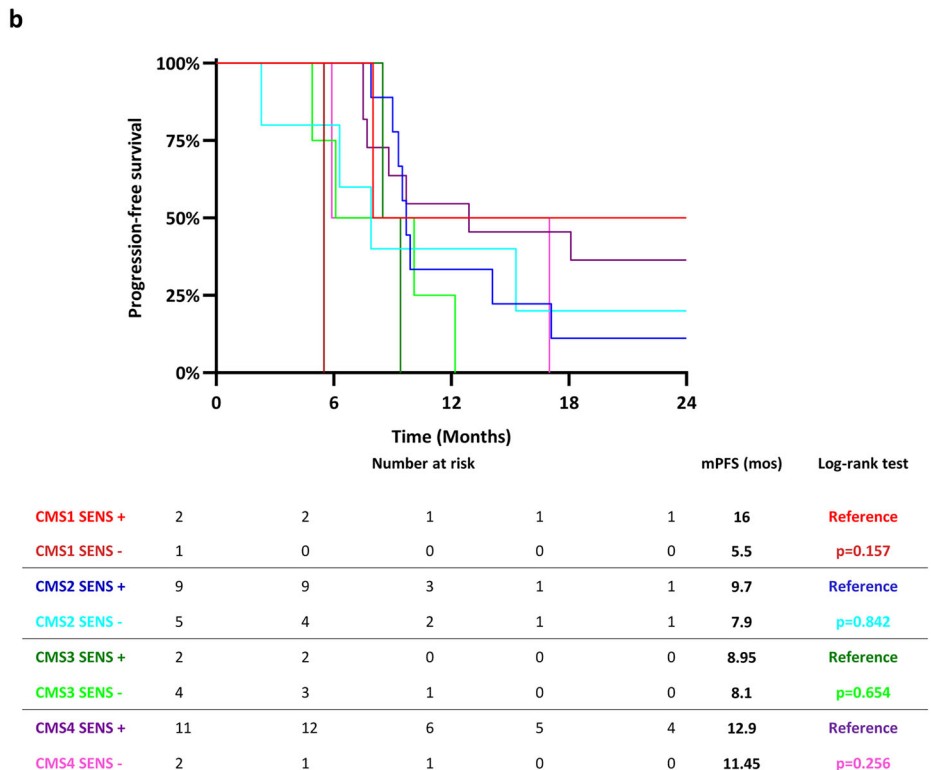

| | Number at risk | | | | | mPFS (mos) | Log-rank test |
|---|---|---|---|---|---|---|---|
| CMS1 SENS + | 2 | 2 | 1 | 1 | 1 | 16 | Reference |
| CMS1 SENS - | 1 | 0 | 0 | 0 | 0 | 5.5 | p=0.157 |
| CMS2 SENS + | 9 | 9 | 3 | 1 | 1 | 9.7 | Reference |
| CMS2 SENS - | 5 | 4 | 2 | 1 | 1 | 7.9 | p=0.842 |
| CMS3 SENS + | 2 | 2 | 0 | 0 | 0 | 8.95 | Reference |
| CMS3 SENS - | 4 | 3 | 1 | 0 | 0 | 8.1 | p=0.654 |
| CMS4 SENS + | 11 | 12 | 6 | 5 | 4 | 12.9 | Reference |
| CMS4 SENS - | 2 | 1 | 1 | 0 | 0 | 11.45 | p=0.256 |

**Fig. 4 | Combination of RES and SENS signatures with Consensus Molecular Subtypes classification. a**, **b** Kaplan-Meier curves of PFS for patients stratified in the CMS subtypes based on RNA-Seq data and for the presence (+) or the absence (—) of genomic alterations in genes belonging to the RES (**a**) and SENS (**b**) signatures. *P* values were calculated by the two-sided Log-rank (Mantel-Cox) test. mPFS median Progression-free Survival; CMS Consensus Molecular Subtype.

**a**

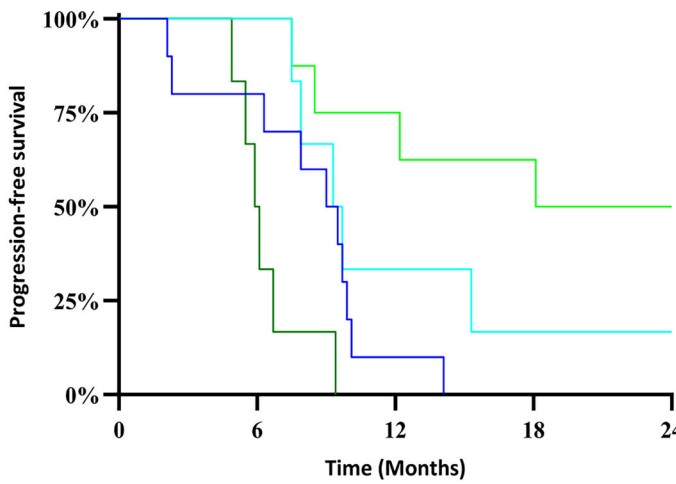

| | Number at risk | | | | mPFS (mos) | Log-rank test |
|---|---|---|---|---|---|---|
| **iCMS2 RES +** | 10 | 8 | 1 | 0 | 0 | **9.25** | Reference |
| **iCMS2 RES -** | 6 | 6 | 2 | 1 | 1 | **9.5** | **p=0.235** |
| **iCMS3 RES +** | 6 | 3 | 0 | 0 | 0 | **6.0** | Reference |
| **iCMS3 RES -** | 8 | 8 | 6 | 5 | 4 | **21.05** | **p=0.0007** |

**b**

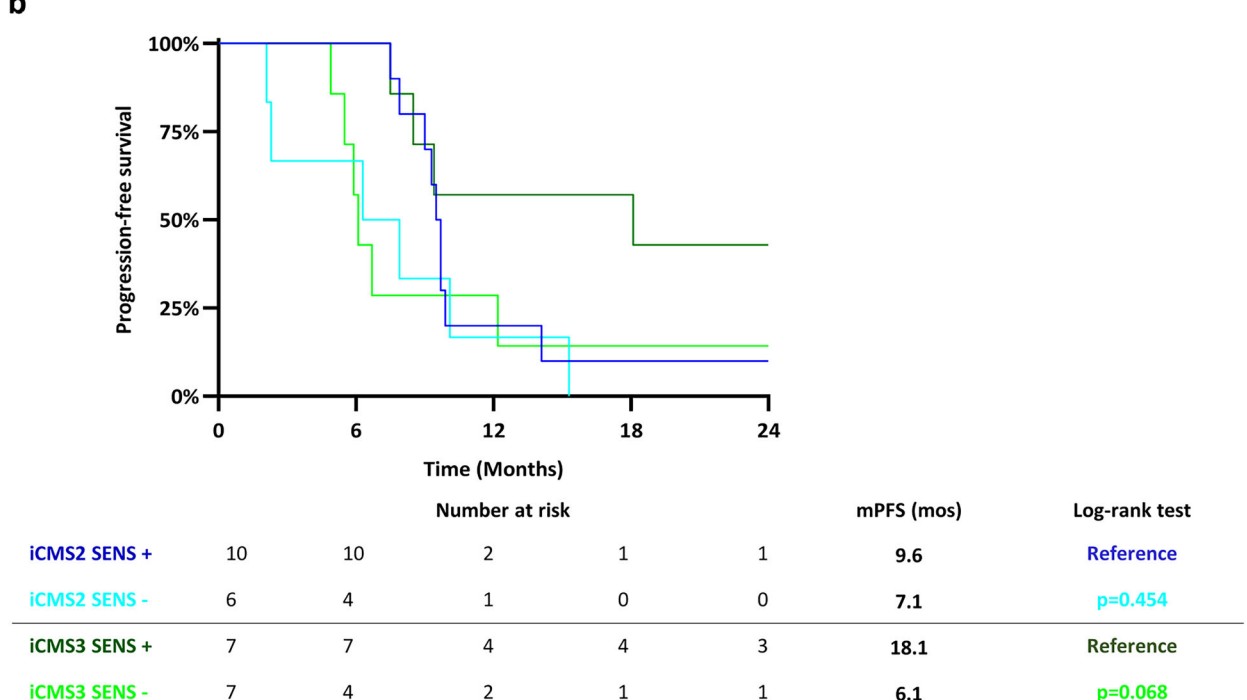

| | Number at risk | | | | mPFS (mos) | Log-rank test |
|---|---|---|---|---|---|---|
| **iCMS2 SENS +** | 10 | 10 | 2 | 1 | 1 | **9.6** | Reference |
| **iCMS2 SENS -** | 6 | 4 | 1 | 0 | 0 | **7.1** | **p=0.454** |
| **iCMS3 SENS +** | 7 | 7 | 4 | 4 | 3 | **18.1** | Reference |
| **iCMS3 SENS -** | 7 | 4 | 2 | 1 | 1 | **6.1** | **p=0.068** |

**Fig. 5 | Combination of RES and SENS signatures with Intrinsic Consensus Molecular Subtypes classification. a**, **b** Kaplan-Meier curves of PFS for patients stratified in the iCMS subtypes based on RNA-Seq data and for the presence (+) or the absence (−) of genomic alterations in genes belonging to the RES (**a**) and SENS (**b**) signatures. *P* values were calculated by the two-sided Log-rank (Mantel-Cox) test. mPFS: median Progression-free Survival; iCMS Intrinsic Consensus Molecular Subtype.

relevant because they were obtained in a cohort of patients mutated for RAS or BRAF and, therefore, with a worse prognosis[21]. The ORR in the AtezoTribe study that enrolled patients with also RAS/BRAF wild-type mCRC was 59%. However, in the NIVACOR trial a higher percentage of dMMR/MSI patients (15% of the total population) was enrolled as compared to the AtezoTribe (5.9%). This phenomenon may have positively impacted the response rate. In addition, we enrolled a quite high fraction of BRAF mutant tumors (16.2%), which showed a higher benefit also in the AtezoTribe trial that included only 8% of BRAF mutant cases in the experimental arm. Finally, we performed an Independent Radiological Central Review which could have made the assessment of the best response more accurate and homogenous. The median PFS in the NIVACOR trial was 10.1 months. We must emphasize that our trial is a phase II study designed to evaluate the objective response rate as primary end-point, with PFS as secondary end-point. Compared to other phase II trials, we do not have adequate statistical power to assess survival. In addition, the different chemotherapy backbone and the different enrolled populations (only RAS/BRAF mutant in NIVACOR versus all comers in other studies) make it difficult to compare PFS with other trials[13,22,23].

The safety profile observed in the trial aligns with previous studies of FOLFOXIRI-based regimens and immune checkpoint inhibitors, with manageable rates of TrAEs. However, the high incidence of irAEs, particularly hypothyroidism and gastrointestinal toxicities, highlights the need for vigilant monitoring and the development of strategies to mitigate treatment-related complications. A notable strength of our study lies in its integration of CGP and transcriptomic analyses, which identified distinct molecular pathways associated with treatment outcomes. Interestingly, patients with high TMB within the pMMR/MSS cohort exhibited prolonged mPFS compared to their TMB-low counterparts, raising questions about the role of TMB as a predictive biomarker in this population. Although the results did not reach statistical significance, they are in line with previous findings from the AtezoTribe study, in which patients bearing TMB-high tumors derived higher benefit from the addition of atezolizumab to FOLFOXIRI-bevacizumab[13,14].

CGP, RNAseq and PPI analyses pointed out to increased activation of PI3K signaling in patients with longer PFS within the NIVACOR MSS cohort. Several mechanisms might be involved in the increased sensitivity to immunotherapy of CRC with mutations in the PI3K pathway[24]. Previous studies suggested that *PIK3CA* mutated CRC have higher TMB levels as compared with non-mutated cancers[25,26]. In this respect, we found a significant correlation between GAs in the PI3K signaling pathways and TMB-high in the whole population and a trend in the MSS cohort. In addition, the loss of PTEN as well as the presence of *PIK3CA* mutations has shown correlations with an increase in the levels of expression of PD-L1 in CRC tumor tissue[27]. However, we did not find a significant difference in the levels of expression of PD-L1 (CD274) in cases with or without GA in the PI3K pathway (Supplementary Fig. 5c).

Overall, these findings further support the rationale for combinations of ICIs with PI3K signaling inhibitors in MSS CRC[28].

The mutational profile of patients with shorter PFS (RES signature) showed a correlation with DNA repair mechanisms. In addition, RNAseq and PPI analyses demonstrated a down-regulation of the double strand-break repair system in cases with longer PFS, as compared with patients with worse prognosis. An alteration of DNA repair mechanisms may determine an increased sensitivity to FOLFOXIRI/bevacizumab/nivolumab therapy through different mechanisms. A deficiency of the double strand-break repair system leads to an increased sensitivity to platinum and its derivatives due to the inability to repair drug-induced DNA damage[29]. Furthermore, an increased genomic instability is associated with an accumulation of genomic alterations with a possible increase of TMB and neoantigens, in addition to effects on the tumor microenvironment, including increased infiltration by immune cells and expression of immune checkpoint molecules[30].

The presence of high levels of TMB and neoantigens is not a sufficient condition for response to ICIs. In fact, an adequate infiltration and activation of immune cells is also necessary for a response[31]. Among patients in the 75% percentile of PFS, 4/9 have high TMB, compared to 0/11 in the 25% percentile. However, all patients in the 75% percentile of PFS present an activation of different biological processes related to the immune response, such as chemotaxis of neutrophils, antigen receptor-mediated signaling and chemokines signaling. Our findings are in agreement with previous observations from the AtezoTribe showing that patients with high Immunoscore IC were more likely to receive higher benefit from the addition of atezolizumab[13,14]. It is important to emphasize that Immunoscore IC and RNAseq are not routine and standardized approaches for CRC profiling. With the increasing use of CGP for the molecular characterization of tumors, having identified possible genomic signatures associated with sensitivity/resistance to ICI therapy in CRC could favor the clinical implementation of biomarkers to select patients who could have greater benefit from this therapeutic approach.

The CheckMate 9×8 study suggested a possible correlation between CMS3 subtype and improved outcomes in patients with mCRC treated with chemotherapy plus nivolumab[22]. This correlation was not confirmed in the NIVACOR patient cohort. However, we must underline that there are numerous differences in the therapy administered and in the population enrolled in the two studies. In CheckMate 9×8, patients received standard therapy with FOLFOX6 plus bevacizumab to which nivolumab was added, while the chemotherapy backbone in NIVACOR is FOLFOXIRI. Furthermore, while NIVACOR enrolled only RAS or BRAF-mutated patients, 57% of patients in CheckMate 9×8 were wild-type. In the CMS3 subgroup, the RES and SENS signatures appear to contribute to better patient stratification. However, the numbers in the different subgroups are too small to allow any conclusions to be drawn.

Two intrinsic subtypes, iCMS2 and iCMS3, have recently been proposed, identified from single-cell sequencing data[18]. The iCMS3 subtype includes both MSI and MSS tumors with high activation of signatures associated with inflammation and immune cell activation. In NIVACOR patients, we found no differences in mPFS between iCMS2 and iCMS3 patients, although after 12 months the curves differed in favor of the iCMS3 group. In patients with an iCMS3 expression profile, both the RES and SENS signatures identified patients with different prognoses. However, we must emphasize that in the original classification, the iCMS3 subtype is enriched in KRAS and BRAF mutations compared to iCMS2. The NIVACOR cohort consists entirely of patients with KRAS and BRAF mutations and this may have introduced a bias in the sub-classification in iCMS, as also suggested by the fact that we could not classify in the iCMS classes 14/48 tumors.

Despite its strengths, the trial has limitations that warrant consideration. The single-arm design and relatively small cohort size limit the generalizability of the findings and the ability to draw definitive conclusions about the efficacy of this combination compared to standard therapies. Additionally, while the translational analyses provided valuable insights, the limited quality of RNA from archival tissue samples constrained the depth of transcriptomic investigations. The limited number of samples available for translational analysis may have determined bias and false positives in our analysis, which should be considered exploratory and requires validation in independent cohorts with similar clinicopathological characteristics. However, the fact that we have identified similar pathways through both DNA and RNA analysis gives us confidence that the biomarkers and pathways identified in this project are worthy of future exploration.

In conclusion, the NIVACOR trial demonstrates the potential of combining immune checkpoint inhibition with intensive chemotherapy and anti-angiogenic therapy in a challenging mCRC population. The integration of molecular profiling into clinical practice may improve patient selection and optimize treatment efficacy. Future randomized studies with larger cohorts and robust biomarker validation are necessary to confirm these findings and establish the role of this combination in the treatment landscape of mCRC.

## Methods

### Study design and participants

The NIVACOR trial is a prospective, open-label, single-arm, multicenter phase II trial in which unresectable, locally advanced, or RAS or BRAF mutated mCRC patients were enrolled from 11 Italian Cancer Centers to receive nivolumab in combination with FOLFOXIRI/bevacizumab as first-line treatment. Local polymerase chain reaction testing assessed *RAS* and/or *BRAF* mutational status. Eligible patients were age >18 years with previously untreated, histologically proven, locally advanced, unresectable, or metastatic RAS/BRAF mutated CRC. The patients had an Eastern Cooperative Oncology Group performance status (ECOG PS) of 0 or 1, > 1 measurable lesions per Response Evaluation Criteria in Solid Tumors (RECIST) version 1.1 criteria, a predicted life expectancy of > 3 months, and adequate organ function. Patients were excluded if presented these conditions: prior monoclonal antibody and chemotherapy received (except neo/adjuvant treatment at least 12 months before the diagnosis of metastatic disease); radiation therapy within 4 weeks before the study; prior anti-PD-1, programmed death ligand 1 (PD-L1), or PD-L2 antibodies; active autoimmune disease; active infection requiring systemic treatment; active hepatitis B or C virus infection; evidence of bleeding diathesis or coagulopathy; unchecked hypertension and prior hypertensive crisis or hypertensive encephalopathy. The study adhered to Good Clinical Practice guidelines and the Declaration of Helsinki.

The study was approved by the Italian Health Authority – AIFA (Agenzia Italiana del Farmaco) on February 8th, 2019, and by the local ethics committees at each participating institution, and registered on August 28th, 2019 at Clinicaltrials.gov (NCT04072198). All patients provided written informed consent.

### Procedures

Eligible enrolled patients received intravenously (IV) FOLFOXIRI/bevacizumab plus nivolumab every 2 weeks for 8 cycles (induction phase) followed by bevacizumab plus nivolumab every 2 weeks (maintenance phase) until disease progression, unacceptable toxicity, or patient/physician decision. Bevacizumab was administered at a dose of 5 mg/kg every 2 weeks, and nivolumab at a flat dose of 240 mg every 2 weeks. FOLFOXIRI was administered as irinotecan 165 mg/m2 for 60 min, followed by oxaliplatin 85 mg/m2 given concurrently with leucovorin at a dose of 200 mg/m2 for 120 min, followed by 5-fluorouracil 3200 mg/m2 continuous infusion for 48 hours. Treatment delay and dose modification because of toxic effects were allowed and specified in the protocol.

The efficacy was evaluated by computed tomography of the chest, abdomen, and pelvis performed every 8 weeks during study treatment, and then every 3 months for 3 years. An Independent Monitoring Committee assessed the safety data of the first ten patients enrolled. During the protocol's treatment, patients will be followed for safety based on Adverse Event (AE) assessments graded by investigators according to National Cancer Institute Common Terminology Criteria for Adverse Events (CTCAE) version 4.03.

### Outcomes

The primary endpoint was the ORR (the proportion of patients with complete response [CR] or partial response [PR]) as assessed by an Independent Radiological Central Review per RECIST v1.1 criteria.

Secondary endpoints were the duration of response (DoR; time from first documented response until disease progression [PD] or death as a result of any cause, whichever occurred first); disease control rate (DCR; the proportion of patients with CR, PR, and stable disease [SD]); PFS (time from first study treatment to first documented PD or death, whichever occurred first); OS (time from first study treatment to death as a result of any cause); safety and quality of life evaluated by EORTC QLQ-C30 questionnaire.

### Nucleic acids extraction

Genomic DNA (gDNA) was isolated from two 10 μm-thick FFPE tissue sections using the GeneRead DNA FFPE kit (QIAGEN) according to the manufacturer's protocol. The gDNA quantity was assessed with the Qubit dsDNA HS assay kit (Thermo Fisher Scientific) using the Qubit 3.0 Fluorometer (Thermo Fisher Scientific).

RNA was isolated from two 10 μm-thick FFPE tissue sections using the RecoverAll Total Nucleic Acid Isolation kit (Thermo Fisher Scientific) according to the manufacturer's instructions. The RNA quantity was assessed with the Qubit RNA HS assay kit (Thermo Fisher Scientific) using the Qubit 3.0 Fluorometer (Thermo Fisher Scientific).

### DNA sequencing

Oncomine Comprehensive Plus Assay (OCA) (Thermo Fisher Scientific) covers a region of 1.50 M total bases, of which 1.06 M exonic bases that include more than 500 genes associated with cancer and is also able to assess complex biomarkers such as TMB and MSI. Libraries were prepared using Ion AmpliSeq Library Kit Plus (Thermo Fisher Scientific) starting from 20 ng of gDNA. Forty picomoles of each library were multiplexed and clonally amplified by emulsion PCR, performed and enriched on the Ion Chef instrument (Thermo Fisher Scientific). Finally, the template was loaded on an Ion 550 Chip and sequenced on an Ion S5 XL sequencer (Thermo Fisher Scientific) according to the manufacturer's instructions. Each sequenced chip contained four samples. The analysis was performed with Ion Reporter v5.20 software by Oncomine Comprehensive Plus −w3.1 -DNA- Single Sample workflow (Thermo Fisher Scientific).

### RNA sequencing

The Ion AmpliSeq Transcriptome Human Gene Expression Kit (THGE) (Thermo Fisher Scientific) was used for targeted RNA sequencing. Using 100 ng of RNA, cDNA was generated using the SuperScript VILO cDNA Synthesis Kit (Thermo Fisher Scientific). Libraries were prepared using Ion AmpliSeq Transcriptome Human Gene Expression kit (Thermo Fisher Scientific). Forty picomoles of each library were multiplexed and clonally amplified by emulsion PCR, performed and enriched on the Ion Chef instrument (Thermo Fisher Scientific). The template was loaded on an Ion 540 Chip and sequenced on an Ion S5 XL sequencer (Thermo Fisher Scientific) according to the manufacturer's instructions. Finally, the BAM files were generated using the AmpliSeqRNA (target region: hg19_AmpliSeq_Transcriptome_21K_v1) plugin.

### RNA sequencing data processing and analysis

RNA sequencing data were further aligned with the CLC Genomics Workbench software v.21.0.1 (Qiagen) and the RNAseq Analysis tool was used to obtain the matrix count for all samples. From the entire gene list, by Qlucore Omics Explorer data analysis software v3.9 (Qlucore), we filtered out low-expressed genes. After removal of genes with count=0 in all samples and of low expressed genes (< 20 counts), the final count matrix included 18153 genes. Next the filtered count matrix was normalized using the TMM method, and principal component analysis (PCA) was done to visualize the data set and better discriminate patients, taking into account PFS. The differentially expressed genes were selected with fold change >2.5 or fold change

<-2.5 using two group statics analysis (t-test) with statistical significance (*p* value = 0.0128). Gene Ontology and Pathways enrichment analyses were performed using the Enrichr tool (https://maayanlab.cloud/Enrichr). The CMS classification was performed processing the expression matrix with the R package CMScaller v2.0.1. Samples were considered not classified if the permutation-based FDR was ≥ 0.05. The iCMS classification was obtained from 715 marker genes of intrinsic epithelial cancer signature previously described[18]. The iCMS2 genes were obtained from the list of genes upregulated in the iCMS2 and downregulated in iCMS3, whereas the iCMS3 genes from the list of genes upregulated in the iCMS3 and downregulated in iCMS2. The classification was performed using the 'ntp' function of the CMScaller R package and samples were defined as not classified if permutation-based FDR was ≥ 0.05.

### Statistical analysis

The baseline characteristics of patients were summarized using absolute numbers and relative frequencies for categorical variables and as mean values ± standard deviation for numerical variables. Comparison among mutational and microsatellite status was respectively performed through the chi-squared test and t-test for independent samples.

The primary objective of the study was evaluated according to the A'Hern modification of the original Fleming one-stage design in which at least 56 responses out of 73 (comprehensive of 10% of drop-out) patients enrolled were necessary to declare the clinical activity of nivolumab added to the standard colorectal first-line chemotherapy not rejecting the alternative hypothesis of increasing the ORR rate from 0.66 to 0.80 with a one-sided alpha error of 5% and 80% of statistical power.

ORR, DCR, and safety variables were reported in terms of absolute and relative frequencies, together with their Clopper Pearson 95% confidence limits.

Swimmer plots were used to graphically show for each patient the induction, maintenance, and follow-up period (blue, light blue, and white bars) enriched by the time occurrence of at least one type of response of disease (CR, PR or SD), PD and Death.

The survival endpoints during the follow-up were graphically depicted using Kaplan-Meier curves. The time to the first endpoint was analyzed for its dependence on the putative predictors using Cox proportional hazard models. All the tests were two-sided at a significance level of 0.05. The statistical analyses were made using R Statistical software, version 4.2.2 (R Foundation for Statistical Computing, Vienna, Austria) and Graph Pad Prism v.10 (Dotmatics).

### Reporting summary

Further information on research design is available in the Nature Portfolio Reporting Summary linked to this article.

## Data availability

Data supporting the findings of this study are available within the article. To protect the privacy of the patients in the study, clinical information of individual RAS and BRAF mutated mCRC patients have been anonymized and reported in Fig. 1, Supplementary Fig. 2 and in the Supplementary Data file. Source data and further clinical data can be requested to the corresponding author within the limitations of the patient informed consent. De-identified sequencing data have been deposited in the Genome Sequencing Archive (accession code HRA016779). The access is restricted and requires approval by the NGDC Data Access Committee upon request to the corresponding author. The accession request will be reviewed within a timeframe of 2-4 weeks and data will be available in accordance with participant consent and applicable regulations. Sequencing data will be also available in the Zenodo repository (https://doi.org/10.5281/zenodo.17358523).

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

## Acknowledgements

This was a nonprofit study, conducted in the GOIRC clinical research network. Bristol Myers Squibb partially covered the costs of the study. The study was also supported by Ricerca Corrente grant L3/8 from Ministero della Salute to REA. NN have received support from Ministero della Salute (Project T3-AN-06 "Sviluppo di una piattaforma per la implementazione clinica della oncologia di precisione nelle regioni del centro-sud Italia"). We thank all patients and their families, all caregivers, the referring oncologists and the GOIRC group for the participating centers.

## Author contributions

Conceptualization: F.I., N.N., C.P. and A.D. Data Curation: S.T., A.D. and G.M. Investigation: R.E.A., S.T., F.B., L.A., G.N., F.P., G.T., T.L., R.B., G.R., E.G., M.L., E.M., S.L. and A.R. Validation: R.E.A. and S.T. Resources: D.F., M.R.M. and A.D.L. Formal analysis: R.E.A., D.R., and G.M. Funding acquisition: C.P. and A.D. Supervision: N.N., C.P. and A.D. Writing – original draft: R.E.A., N.N., A.D. and G.M. Writing–review & editing All authors.

## Competing interests

Angela Damato: outside the submitted work, has received personal fees for the advisory role, speaker engagements, and travel and accommodation expenses from Ipsen, Servier, BMS, Merck Serono, Amgen, and Daiichi Sankyo. Francesca Bergamo: received personal honoraria as invited speaker from Eli-Lilly, MSD, Bristol Myers Squibb, AstraZeneca, Bayer, Pierre Fabre; participation in advisory board for AAA Novartis, Takeda, Teysuno. Filippo Pietrantonio: honoraria: Servier, Bayer, AstraZeneca, Lilly, MSD, Amgen, Pierre-Fabre, Merck-Serono, BMS, Astellas. Consulting or Advisory Role: Merck-Serono, Amgen, Servier, MSD, Organon, Bayer. Research Funding: Bristol Myers Squibb (Inst), AstraZeneca (Inst), Incyte, Agenus. Giuseppe Tonini: advisory board member of Molteni, MSD, Novartis, Roche, and Pharmamar. Elisa Giommoni: advisory board of Eli Lilly, Amgen, Viatris, BMS, Servier. Sara Leonardi: received honoraria (as invited speaker) from Roche, Eli Lilly, BMS, Servier, Merck Serono, Pierre Fabre, GSK, and Amgen; consulting fee (advisory boards) from Amgen, Astellas, Bayer, Merck Serono, Eli Lilly, AstraZeneca, Incyte, Daiichi-Sankyo, BMS, Servier, Merck Sharp & Dohme (MSD), GSK, Takeda, Rottapharm, and Beigene; as well as grants or funds (to institution) from Amgen, Merck Serono, Bayer, Roche, Eli Lilly, AstraZeneca, and BMS. Nicola Normanno: outside the submitted work personal fees for the advisory role, speaker engagements, and travel and accommodation expenses from AstraZeneca, Bayer, Biocartis, Bristol Myers Squibb, Eli Lilly, Illumina, Incyte, MERCK, Merck Sharp & Dohme, Novartis, Roche, Servier, Thermofisher; financial support to research projects from Astrazeneca, Biocartis, Illumina, MERCK, QIAGEN, Roche, Sophia Genetics, Thermofisher. Carmine Pinto: outside the submitted work personal fees for the advisory role, speaker engagements, and travel and accommodation expenses from Amgen, Astellas, AstraZeneca, Bayer, Bristol Meyer Squibb, Celgene, Clovis Oncology, Eisai, Ipsen, Janssen, Incyte, Merck-Serono, Merck Sharp and Dohme, Novartis, Roche, Sandoz, Sanofi, and Servier. The other authors have no conflicts of interest to declare.

## Additional information

Angela Damato [1,16], Riziero Esposito Abate[2,16], Simona Tessitore[2], Daniela Frezzetti[2], Monica Rosaria Maiello[2],
Dario Righelli [3], Francesca Bergamo[4], Lorenzo Antonuzzo[5,6], Guglielmo Nasti[7], Filippo Pietrantonio[8], Giuseppe Tonini[9,10],
Tiziana Latiano[11], Roberto Bordonaro[12], Gerardo Rosati[13], Elisa Giommoni [6], Francesco Iachetta[1], Mario Larocca[1],
Evaristo Maiello[11], Sara Lonardi [4], Alessandra Romagnani [1], Giuseppe Maglietta[14], Antonella De Luca[2],
Nicola Normanno [15] ✉ & Carmine Pinto[1]

[1]Medical Oncology, Comprehensive Cancer Centre, AUSL-IRCCS di Reggio Emilia, Reggio Emilia, Italy. [2]Cell Biology and Biotherapy Unit, Istituto Nazionale
Tumori Fondazione G. Pascale – IRCCS, Naples, Italy. [3]Department of Electrical engineering and information technologies, University Hospital "Federico II" of
Naples, Naples, Italy. [4]Medical Oncology 1, Veneto Institute of Oncology IOV - IRCCS, Padua, Italy. [5]Department of Experimental and Clinical Medicine,
University of Florence, Florence, Italy. [6]Oncology Unit, Careggi University Hospital, Florence, Italy. [7]Abdominal Oncology Division, Istituto Nazionale Tumori
Fondazione G. Pascale – IRCCS, Naples, Italy. [8]Medical Oncology Department, Fondazione IRCCS Istituto Nazionale dei Tumori, Milan, Italy. [9]Medical
Oncology, Fondazione Policlinico Universitario Campus Bio-Medico, Rome, Italy. [10]Department of Medicine and Surgery, Università Campus Bio-Medico di
Roma, Rome, Italy. [11]Oncology Unit, Foundation IRCCS, Casa Sollievo della Sofferenza, San Giovanni Rotondo, Italy. [12]Medical Oncology Unit, ARNAS
Garibaldi, Catania, Italy. [13]Medical Oncology Unit, San Carlo Hospital, Potenza, Italy. [14]Clinical and Epidemiological Research Unit, University Hospital of
Parma, Parma, Italy. [15]Scientific Directorate, IRCCS Istituto Romagnolo per lo Studio dei Tumori (IRST) "Dino Amadori", Meldola, Italy. [16]These authors
contributed equally: Angela Damato, Riziero Esposito Abate. ✉e-mail: nicola.normanno@irst.emr.it; nicnorm@yahoo.com

