## [Transparent Peer Review file · Nature Communications]

First-line Nivolumab plus FOLFOXIRI/Bevacizumab in advanced RAS/BRAF-mutated colorectal cancer: efficacy, safety and biomarker discovery from the phase II NIVACOR trial.

Corresponding Author: Dr Nicola Normanno

Version 0:

Reviewer comments:

Reviewer #2

(Remarks to the Author)

Immune therapy has shown strong efficacy in dMMR/MSI metastatic colorectal cancer (mCRC) but limited effect in pMMR/MSS mCRC patients. This study presents the results of a phase II NIVACOR trial, the combination of FOLFOXIRI/bevacizumab with nivolumab was evaluated as a first-line treatment for RAS/BRAF-mutated mCRC. Among 73 patients, 76.7% achieved objective responses, and the disease control rate was 97.3%. The median progression-free survival (mPFS) was 10.1 months, with the median overall survival (mOS) not yet reached. Further genomic and RNA sequencing identified pathways, including PI3K/AKT, DNA repair, and T cell activation, associated with treatment response. These results suggest a potential synergy between immune checkpoint inhibitors and chemotherapy in pMMR/MSS mCRC patients. Overall, this Phase II NIVACOR trial demonstrated that the combination of FOLFOXIRI/bevacizumab with nivolumab as first-line treatment in RAS/BRAF-mutated pMMR/MSS mCRC showed promising objective response and disease control rates, but further investigation is needed to fully understand its long-term efficacy and underlying mechanisms. The title of the article emphasizes "Comprehensive Genomic Profiling", but the corresponding DNA sequencing data and RNA sequencing analysis are very superficial, with only RNA-seq comparing the differential genes of patients in the upper and lower quartiles of PFS. How to effectively connect genomic profiling and trial has not been solved.

Major comments

1. Patient recruitment ended on March 4, 2021, more than four years ago, and the authors were able to calculate the corresponding overall survival.
2. Do different RAS mutations affect the efficacy of the combination of FOLFOXIRI/bevacizumab with nivolumab?
3. Supplementary Table 1 provides baseline demographics and disease characteristics. The authors should provide detailed information for each patient, such as age, gender, stage, PFS, OS, treatment information, etc.
4. Why is TMB >10.4 mutations/Mb defined as TMB-high? How is the mPFS of MSI patients compared with MSS-TMB-high and MSS-TMB-low patients?
5. It is very unclear how to define a sensitivity signature (SENS).
6. It is not clear how many samples were used for RNA sequencing? Why compare DEGs between 75th and 25th percentile PFS patients? The corresponding figure lacks figure annotation.

Minor comments

1. The authors need to label the sample size and p value of statistical analysis in Figure 2 and 3.
2. Line 197 "Enrichr analysis" should be typo, please confirm it.
3. Detailed RNA-seq analysis should be described in Method section but not in main context.

Reviewer #3

(Remarks to the Author)

The paper by D'Amato et al summarizes clinical and translational findings from a phase II single arm trial, named NIVACOR, investigating the safety and activity of FOLFOXIRI+ bevacizumab + nivolumab in the upfront therapy of RAS or BRAF mut mCRC.

ORR results are very promising while PFS data are less convincing.

Overall, the single arm nature of the trial is a clear limitation to understand the added value of the anti-PD1 on top of the standard-of-care (for selected pts) regimen.

A big limitation is the inclusion of dMMR mCRC pts and the high percentage of dMMR pts enrolled, likely due to the willingness of treating physicians to offer them ICIs if not available in clinical practice. As a consequence the number of pts where the scientific question behind the study is still clinically relevant is even more limited.

Translational analyses do not help a lot in increasing the scientific value of the paper, since the approach is like a comprehensive molecular characterization by DNA and RNA in a "wide fishing" way rather than an attempt to verify specific hypotheses. As a consequence, the risk of falsely positive findings is very high, the probability of having chance-driven results is relevant, and this bias is minimally discussed.

The lack of any validation of signatures built on these samples in independent cohorts, knowing the prognostic group to which they belong, is a significant limitation for all the translational findings. Multiple mRNA signatures or phenotypic markers of benefit from ICIs in pMMR mCRC have been suggested, and demonstrating their utility or not would have been more useful in my opinion than searching for other markers without any chance to demonstrate their predictive rather than prognostic meaning.

Reviewer #4

(Remarks to the Author)

The statistical and bioinformatics plan is well described in the Method, such as the use of A'Hern modification of the Fleming one-stage design to determine the minimum number of responders (at least 56 responses out of 73 patients) to claim success for testing the primary hypothesis: 80% vs 66% ORR. The data analysis for the clinical trial was generally reasonable; however, additional detail is needed—particularly regarding the analysis of both pMMR/MSS and dMMR/MSI subgroups. While the response rate appears impressive, it remains unclear whether the progression-free survival (PFS) results are clinically meaningful. Furthermore, several issues were identified, especially in the translational analysis that warrant clarification and improvement as listed below.

1. Enhancement is needed in introductory content. The literature review should include key statistics of the cited papers, such as response rate and median PFS, for mCRC, and the subgroups, pMMR/MSS and dMMR/MSI, to contrast the study results.
2. The result is quite impressive in improving ORR to 70-78% in both subgroups (RAS/BRAF mutated mCRC with poor prognosis) compared to AtezoTribe study with 59% ORR in RAS/BRAF wild-type mCRC. Please comment on why and how nivolumab improves this population. Also, did the PFS data show similar findings?
3. What is the rate for the overall treatment-related (Tr) AEs and also for Grade 3-5-specific? Was the rate of treatment-related AEs higher than non-treatment-related AEs? Also what were the AE rates (Tr AE and irAE) for the two subgroups (pMMR/MSS and dMMR/MSI)?
4. Questionable translational analysis using Q1 and Q3 PFS. It is unclear how the 25th percentile (Q1) (poor prognosis subgroup) or in 75th percentile (Q3) (good prognosis subgroup) of the PFS values were defined given some patients with censored status. For example, if a patient had a PFS censored in the 1st month, the patient may have a very long PFS if the patient is able to be followed up. In other words, grouping based on PFS value could be biased without considering censored status.

Another major question is a classification based on "resistance signature" (RES), which was composed of 53 genes altered only in the 25th percentile subgroup. If I interpret it correctly, patients with RES would be only in PFS Q1 group. Accordingly, the survival plot was just a similar comparison between Q1 vs Q2-Q3, not due to RES. Moreover, when patients in the Q1 group were excluded from the analysis, it means the remaining patients should be RES wild-type. How was the analysis conducted in comparison with the mutant vs wildtype. The same concern applies to the Q3 PFS group for the sensitivity signature (SENS). If these issues are not addressed, the remaining translational analysis could be problematic.

5. In Figure 5, it is unclear why there are only 19 patients presented to highlight the difference between the 75th vs 25th PFS comparison. The author indicated that 48 samples were profiled with RNA sequencing – so the number should be around 24.
6. The need for a comparison with published signatures. Since the RES and SENS appear to be a major point in the study, I would recommend presenting a supplementary table of these genes. The gene symbols are hard to see in supplementary figure S6. A comparison of the sensitivity signature and resistance signature with published signatures associated with primary and metastatic CRC.
7. The need for high-resolution figures. It is difficult to view Fig 1's swimmer plot. Also, the survival plot needs to include number of at risk and event at each time point.
8. Lack of subgroup analysis (pMMR/MSS and dMMR/MSI). While the overall results were presented, it is more informative to include analysis results for both pMMR/MSS and dMMR/MSI subgroups, such as median treatment duration, median follow-up, similar Fig 1 for each subgroup, and others.
9. Since this is a clinical trial report, baseline demographic and disease characteristics should be in the primary table, not in the supplementary table.
10. The study indicates the population includes RAS/BRAF mutated mCRC in 73 patients. However, from supplementary figure 3, there are ~50 KRAS and ~10 BRAF mutations, which don't add up.
11. The bioinformatics method that was applied for the differential gene expression analysis should be more clearly stated, such as if the author applied DEseq2.
12. Based on the presented analysis, the user should refrain from making inferences related to lymphocyte-T CD8+ and

regulation of T-reg cells without direct measurements related to these cells.

Reviewer #5

(Remarks to the Author)

Version 1:

Reviewer comments:

Reviewer #2

(Remarks to the Author)

The authors have addressed the reviewer's comments, with significant improvements in data transparency, integration of clinical and genomic analyses, and methodological clarity. The reviewer has not additional comments.

Reviewer #3

(Remarks to the Author)

I appreciate the efforts by authors in implementing especially the translational part of the manuscript and adding pieces of info useful to advance research in the field.

Reviewer #4

(Remarks to the Author)

The authors have addressed most of our concerns, including providing a detailed literature review and safety analysis. However, concerns remain regarding the RES and SENS signatures due to potential selection bias:

- RES Signature: "To establish the RES signature, we selected a group of 51 genes altered in the subgroup of patients with poor prognosis (12 patients with PFS below the 25th percentile – Q1 PFS) and not altered in the subgroup of patients with good prognosis (12 patients with PFS above the 75th percentile – Q4 PFS)."
- SENS Signature: "We selected a group of 102 genes altered in the good prognosis subgroup (Q4 PFS) and not altered in the poor prognosis subgroup (Q1 PFS)."

Both signatures were derived by comparing Q1 (n=12) and Q4 (n=12) PFS groups, leaving approximately 44 patients untested. A more unbiased approach would be to statistically evaluate these remaining patients (n≈44) based on RES and SENS classifications. Figures 2B and 2C may reflect biased results, as they likely include most Q1 patients in the RES+ group and Q4 patients in the SENS+ group, potentially inflating statistical significance. We recommend redoing the survival analysis by excluding Q1 and Q4 groups to reduce bias.

A similar concern applies to the stratification analysis of RES and SENS within CMS and iCMS subtypes. If Q1 patients disproportionately contribute to the RES+ group in CMS2 and iCMS3, this could also lead to biased statistical significance.

Reviewer #5

(Remarks to the Author)

Version 2:

Reviewer comments:

Reviewer #4

(Remarks to the Author)

The authors have included additional analysis by excluding Q1 for unbiased analysis to show similar results (Same analysis for excluding Q4). No further comment.

Reviewer #5

(Remarks to the Author)

Before responding in detail to the reviewers' remarks, we would like to thank the reviewers for their comments, which have helped us improve our paper. We have written this sentence several times, but in this case, the thanks are truly sincere. The reviewers' observations prompted us to update our patient follow-up and analyze our data with previously published prognostic signatures for colorectal cancer. These analyses strengthened our conclusions and we believe they have improved our paper overall. Therefore, we thank the reviewers and editors for taking the time to provide appropriate and helpful suggestions.

REVIEWER COMMENTS

Reviewer #2 (Remarks to the Author): computational expertise in CRC multi-omics

Immune therapy has shown strong efficacy in dMMR/MSI metastatic colorectal cancer (mCRC) but limited effect in pMMR/MSS mCRC patients. This study present the results of a phase II NIVACOR trial, the combination of FOLFOXIRI/bevacizumab with nivolumab was evaluated as a first-line treatment for RAS/BRAF-mutated mCRC. Among 73 patients, 76.7% achieved objective responses, and the disease control rate was 97.3%. The median progression-free survival (mPFS) was 10.1 months, with the median overall survival (mOS) not yet reached. Further genomic and RNA sequencing identified pathways, including PI3K/AKT, DNA repair, and T cell activation, associated with treatment response. These results suggest a potential synergy between immune checkpoint inhibitors and chemotherapy in pMMR/MSS mCRC patients. Overall, this Phase II NIVACOR trial demonstrated that the combination of FOLFOXIRI/bevacizumab with nivolumab as first-line treatment in RAS/BRAF-mutated pMMR/MSS mCRC showed promising objective response and disease control rates, but further investigation is needed to fully understand its long-term efficacy and underlying mechanisms. The title of the article emphasizes "Comprehensive Genomic Profiling", but the corresponding DNA sequencing data and RNA sequencing analysis are very superficial, with only RNA-seq comparing the differential genes of patients in the upper and lower quartiles of PFS. How to effectively connect genomic profiling and trial has not been solved.

Major comments

1. Patient recruitment ended on March 4, 2021, more than four years ago, and the authors were able to calculate the corresponding overall survival.

We thank the reviewer for this comment that provides us the opportunity to improve our manuscript.

As stated in the study protocol approved by the Ethics Committee, the maximum follow-up for each enrolled patient was limited to 2 years. We updated the database with the most recent follow-up for patients enrolled in the final months of recruitment. All time-dependent data have been updated accordingly in the relevant results and figures.

The median overall survival (OS) was not reached at the end of the study. Deaths occurred in 28/73 enrolled patients (38.4%). These data and updated OS curves (**Figure 1e**) have been added to the new version of the manuscript.

2. Do different RAS mutations affect the efficacy of the combination of FOLFOXIRI/bevacizumab with nivolumab?

The reviewer raised an interesting point, also in light of recent literature data. Information on the specific RAS mutation of the enrolled patients was not provided by the centers participating in the

study. However, RAS mutational status was determined using NGS sequencing. Information on specific KRAS mutations was available for 47 patients. Single nucleotide variants were detected in KRAS codons 12 (n. 34), 13 (n. 9), and 61 (n. 4). Since individual mutations were poorly represented, we combined G12X and non-G12X mutations. The mPFS of G12X and non-G12X patients was 9.55 and 12.9 months, respectively (p = 0.6). This information has been added to the results (Clinical activity in the overall and subgroup population section) and survival curves are shown in the **Supplementary Figure 1d**.

	Number at risk					mPFS (mos)	Log-rank test
RAS G12X	34	29	14	8	7	9.55	Reference
RAS no G12X	13	10	6	4	4	12.9	p=0.607

3. Supplementary Table 1 provides baseline demographics and disease characteristics. The authors should provide detailed information for each patient, such as age, gender, stage, PFS, OS, treatment information, etc.

We thank the reviewer for raising this relevant issue. In order to accommodate the reviewer's request and also the request of Reviewer 4, point 9, we moved the supplementary table 1 in the main text (**Table 1** in the updated version of the manuscript).

Individual patient data are also provided in a **Supplementary Data** file after complete anonymization to comply with Italian clinical trial regulations and protect patient privacy.

4. Why is TMB >10.4 mutations/Mb defined as TMB-high? How is the mPFS of MSI patients compared with MSS-TMB-high and MSS-TMB-low patients?

We thank the reviewer for giving us the opportunity to provide more details.

We previously defined the TMB cut-off value ≥ 10.4 mut/Mb for the OncoPrint Comprehensive Plus Assay (OCA) by comparing this assay with the reference FoundationOne Test. These findings have been published in the Journal for Immunotherapy of Cancer (10.1136/jitc-2023-007800) and are routinely used as reference by our laboratory and other laboratories working in the field to categorize tumors in high versus low TMB using the OCA.

The mPFS of MSI patients is not reached, whereas the mPFS of the MSS-TMB high is 18.1 months and the mPFS of the MSS-TMB-low is 9.35 months. These data are updated in the text

(Comprehensive Genomic Profiling analysis section) and shown in the figure below that is **Figure 2a** in the revised version of the manuscript.

	Number at risk				mPFS (mos)	Log-rank test	
MSS-Low TMB	40	32	15	7	7	9.35	Reference
MSS-High TMB	9	9	5	5	4	18.1	p=0.083
MSI	6	6	6	5	5	NR	p=0.006

5. It is very unclear how to define a sensitivity signature (SENS).

We agree with the Reviewer that we did not provide a clear definition of the SENS signature.

Comprehensive genomic profiling was successfully completed for 55 cases, including 49 MSS patients. We identified a “SENS signature” by comparing genomic alterations of patients with good prognosis (patients with PFS over the 75th percentile, Q4 PFS) with the poor prognosis subgroup (patients with PFS below the 25th percentile, Q1 PFS) within the cohort of 49 MSS tumors. We found 102 genes that showed alterations only in the good prognosis subgroup but not in the poor prognosis group. We hypothesized that these genomic alterations could be possibly associated with better prognosis and we therefore defined these genes as SENS signature. Alterations in any of the genes included in the SENS signature were also identified in 19 patients of the cohort between the 75th and the 25th percentile (Q2-Q3 PFS). We better clarified the definition of the SENS signature in the Comprehensive Genomic Profiling analysis section.

6. It is not clear how many samples were used for RNA sequencing? Why compare DEGs between 75th and 25th percentile PFS patients? The corresponding figure lacks figure annotation.

We apologize with the Reviewer for the lack of clarity. Forty-eight samples (42 MSS and 6 MSI) were successfully analysed using RNA Seq. Among the 42 MSS patients, 11 were in the 25th PFS percentile subgroup and 9 in the 75th PFS percentile subgroup. We focused our attention on the DEGs between the subgroup of patients with better prognosis (longer PFS, 75th percentile, Q4) and the subgroup of patients with the worse prognosis (shorter PFS, 25th percentile, Q1) in order to identify pathways potentially associated with the activity of the experimental treatment in MSS patients. We better described RNA-Seq analyses in the Transcriptomic analysis section. As suggested by the Reviewer, we integrated the legend of Figure 5 (now **Figure 3b**) with more details.

Minor comments

1. The authors need to label the sample size and p value of statistical analysis in Figure 2 and 3.

We apologize with the Reviewer and included the sample size and the p value in the figures of survival curves in the manuscript, including Figure 2 and 3 (now **Figures 2a-c**).

2. Line 197 "Enrichr analysis" should be typo, please confirm it.

In Line 197 "Enrichr analysis" was replaced with "Enrichment analysis" (now line 273).

3. Detailed RNA-seq analysis should be described in Method section but not in main context.

We thank the reviewer for this suggestion. Details on RNAseq analysis in lines 199-201 have been moved in the "Data Processing and Analysis" in the Methods section.

Reviewer #3 (Remarks to the Author): expertise in CRC clinical trials

The paper by D'Amato et al summarizes clinical and translational findings from a phase II single arm trial, named NIVACOR, investigating the safety and activity of FOLFOXIRI+ bevacizumab + nivolumab in the upfront therapy of RAS or BRAF mut mCRC.

ORR results are very promising while PFS data are less convincing.

We thank the reviewer for this comment. We agree that the mPFS of 10.1 months may appear limited when compared to results from previous studies. However, our trial was not designed to evaluate the survival outcomes, but rather the response to a novel therapeutic approach. As a secondary endpoint, compared to other phase II trials, we do not have adequate statistical power for the PFS endpoint. Moreover, as expected from the study design, our study enrolled only patients with RAS and BRAF mutations, who have a worse prognosis, and this may have influenced mPFS values. For example, in the AtezoTribe study, 16% of patients in the atezolizumab arm were RAS/BRAF wild-type, and 3% had no RAS/BRAF mutation data available.

Overall, the single arm nature of the trial is a clear limitation to understand the added value of the anti-PD1 on top of the standard-of-care (for selected pts) regimen.

We agree in part with the reviewer's comment. This is a phase II trial with objective response rate (ORR) as the primary endpoint. According to current regulatory guidance, single-arm study designs are acceptable in phase II trials aimed at evaluating antitumor activity, particularly when the primary endpoint is not time-to-event.

Specifically:

The U.S. Food and Drug Administration (FDA), in its guidance "Clinical Trial Endpoints for the Approval of Cancer Drugs and Biologics" (finalized December 2018), states that while randomized controlled trials are generally preferred when the primary endpoint is a time-dependent measure such as progression-free survival (PFS) or overall survival (OS), non-randomized, single-arm designs are considered acceptable when the primary endpoint is ORR, especially in exploratory phase II settings or rare populations. (FDA, 2018. Available at: <https://www.fda.gov/media/71195/download>)

The European Medicines Agency (EMA), in its "Guideline on the evaluation of anticancer medicinal products in man" (Revision 6, July 2017), confirms that non-comparative (i.e., single-arm) studies may be justified in phase II when the objective is to assess antitumor activity (e.g., ORR). Randomized trials are instead recommended when the primary objective is based on time-to-event endpoints. (EMA, 2017. EMA/CHMP/205/95 Rev.6. Available at: https://www.ema.europa.eu/en/documents/scientific-guideline/guideline-evaluation-anticancer-medicinal-products-man_en.pdf)

The International Council for Harmonisation (ICH), in its E9 guideline (1998) and E9(R1) addendum (2019), emphasizes that the selection of study design should be guided by the objective of the trial and the nature of the primary endpoint, not solely by the trial phase. (ICH E9(R1), 2019. Available at: https://database.ich.org/sites/default/files/E9-R1_Step4_Guideline_2019_1203.pdf)

As practical examples, the recent AtezoTRIBE trial (Antoniotti et al., *Lancet Oncol.* 2022;23(7):876–87), and the CheckMate 9X8 trial (Lenz H-J et al., *J Immunother Cancer.* 2024;12(3):e008409) are randomized phase II studies in metastatic colorectal cancer. In both cases, randomization was implemented because their primary endpoint was PFS, a time-dependent outcome. These examples further underscore that randomization in phase II is driven by the endpoint, not by the phase itself.

In our study, although PFS is of clinical interest and included as a secondary endpoint, the primary objective is ORR, which — based on previous assessments by regulatory agencies — does not require a randomized design at this phase. We therefore maintain that the chosen trial design is methodologically sound and fully aligned with current international regulatory standards. However, we agree that the efficacy of the combination analyzed in our study should be explored in randomized trials, in which patient selection is possibly guided by biological characteristics of the tumor to maximize the effects.

A big limitation is the inclusion of dMMR mCRC pts and the high percentage of dMMR pts enrolled, likely due to the willingness of treating physicians to offer them ICIs if not available in clinical practice. As a consequence the number of pts where the scientific question behind the study is still clinically relevant is even more limited.

We thank the reviewer for this relevant comment, which nevertheless reflects the current knowledge on which the treatment of patients with mCRC is based. However, the NIVACOR study was conceived and designed in 2018 when no solid data on immunotherapy in the dMMR/MSI population were available. Indeed, the results of the KEYNOTE-177 study were published in 2020, and pembrolizumab was approved for reimbursement by the National Health Service in Italy as of February 2022. Therefore, there was no scientific rationale for excluding the dMMR/MSI population from the study at the time it was designed. Furthermore, we focused our study only on patients with RAS/BRAF mutations and therefore, in general, with a worse prognosis. However, a fraction of BRAF mutant tumors are also dMMR/MSI, and this factor may have contributed to somewhat enriching our cohort of dMMR/MSI cases. We must emphasize that other randomized phase 2 trials in the same setting of disease (e.g., AtezoTribe, Checkmate 9X8) have not selected the population and have not excluded dMMR/MSI tumors.

Finally, the preliminary results of NIVACOR in the dMMR/MSI population nonetheless offer important food for thought. In our study, we observed a 100% DCR in this subgroup, while in dMMR/MSI patients treated with pembrolizumab (Keynote-177) or nivolumab plus ipilimumab (CheckMate-142), the best response was disease progression in 29% and 12% of the cases, respectively. These data suggest that the dMMR/MSI population is heterogeneous and that some patients may benefit from more aggressive treatments, including chemotherapy. These considerations have been added to the discussion.

Translational analyses do not help a lot in increasing the scientific value of the paper, since the approach is like a comprehensive molecular characterization by DNA and RNA in a "wide fishing" way rather than an attempt to verify specific hypotheses. As a consequence, the risk of falsely positive findings is very high, the probability of having chance-driven results is relevant, and this bias is minimally discussed.

We thank the reviewer for this observation, which offers us the opportunity to further clarify our approach to translational analyses. The search for biomarkers of immune checkpoint inhibitor efficacy in mCRC has not yet yielded consistent results. Our approach, based on CGP and RNAseq, is obviously an exploration that will require future validation. This approach can certainly lead to biases and false positives, and, as suggested by the reviewer, we have emphasized these limitations more fully in the discussion. However, the fact that we have identified similar pathways through both DNA and RNA analysis gives us confidence that the biomarkers and pathways identified in this project are worthy of future exploration.

The lack of any validation of signatures built on these samples in independent cohorts, knowing the prognostic group to which they belong, is a significant limitation for all the translational findings.

Multiple mRNA signatures or phenotypic markers of benefit from ICIs in pMMR mCRC have been suggested, and demonstrating their utility or not would have been more useful in my opinion than searching for other markers without any chance to demonstrate their predictive rather than prognostic meaning.

We thank the reviewer for raising a very important issue. We searched for similar cohorts of patients with mCRC, i.e., those with RAS and/or BRAF mutations, who had received treatment with polychemotherapy and immune checkpoint inhibitors as first-line therapy, and with available CGP and/or RNAseq data to validate our hypothesis. Unfortunately, we did not find any cohorts with available data to evaluate our signatures. Regarding available biomarkers, we evaluated the impact of TMB and, in this new version of the manuscript, that of both the consensus molecular subtypes (CMS) and the intrinsic CMS (iCMS) classifications (please see also reply to point 6 of reviewer 4). We found that our approach leads to a better stratification of patients as compared with both CMS and iCMS. We fully agree that our analyses is somehow preliminary and that larger case series will be essential to confirm these results and refine the signatures. However, we consider the evidence we presented innovative and relevant to guide future research in the field.

Reviewer #4 (Remarks to the Author): expertise in clinical trial biostatistics

The statistical and bioinformatics plan is well described in the Method, such as the use of A'Hern modification of the Fleming one-stage design to determine the minimum number of responders (at least 56 responses out of 73 patients) to claim success for testing the primary hypothesis: 80% vs 66% ORR. The data analysis for the clinical trial was generally reasonable; however, additional detail is needed—particularly regarding the analysis of both pMMR/MSS and dMMR/MSI subgroups. While the response rate appears impressive, it remains unclear whether the progression-free survival (PFS) results are clinically meaningful. Furthermore, several issues were identified, especially in the translational analysis that warrant clarification and improvement as listed below.

1. Enhancement is needed in introductory content. The literature review should include key statistics of the cited papers, such as response rate and median PFS, for mCRC, and the subgroups, pMMR/MSS and dMMR/MSI, to contrast the study results.

We thank the reviewer for the comment. We have added the requested data in the revised version.

2. The result is quite impressive in improving ORR to 70-78% in both subgroups (RAS/BRAF mutated mCRC with poor prognosis) compared to AtezoTribe study with 59% ORR in RAS/BRAF wild-type mCRC. Please comment on why and how nivolumab improves this population. Also, did the PFS data show similar findings?

We thank the reviewer for the comment. The NIVACOR results are somehow unexpected and surprising, especially because we selected a subgroup of patients with the worst prognosis (RAS and BRAF mutated mCRC). However, the ORR in the AtezoTribe is also unexpected. Indeed, the TRIBE study (Loupakis et al. N Engl J Med 2014;371:1609-18) reported an ORR in the experimental arm (FOLFOXIRI plus bevacizumab) of 65.1%. The AtezoTribe trial had a lower ORR (59%) with an intensified regimen (FOLFOXIRI plus bevacizumab and atezolizumab). Interestingly, the ORR in the phase II Checkmate 9X8 trial (Lenz et al., JITC 2023) was 60%, but with a regimen of chemotherapy less intensified (FOLFOX).

These finding can hardly be explained. We provide some hypotheses that have been added to the discussion:

- 1) In our study, a higher percentage of dMMR/MSI patients (15% of the total population) was enrolled as compared to other trials (13 MSI patients out of 218 randomized in the AtezoTribe study, 5.9%). This phenomenon may have positively impacted the response rate.
- 2) We enrolled a quite high fraction of BRAF mutant tumors (16.2%), which showed a higher benefit also in the AtezoTribe trial.
- 3) We performed an independent central review that may have more scrupulously identified the best response.

Regarding survival outcomes, our trial was designed to evaluate the response rate of this combination strategy, with PFS as a secondary end-point. Compared to other phase II trials, we do not have adequate statistical power to assess survival. Additionally, we have enrolled a population with the worst prognosis (RAS and BRAF mut) making comparison with other trials difficult.

For example, in the phase II AtezoTribe trial, benefit with first-line atezolizumab plus FOLFOXIRI/BEV versus FOLFOXIRI/BEV alone was observed (mPFS = 13.1 months vs 11.5

months; HR, 0.69 (80% CI, 0.56 to 0.85); $p=0.012$). However, 16% of the patients enrolled in the experimental arm were RAS/BRAF wild type and in 3% of the patients the data are missing. The phase II Checkmate 9X8 trial did not demonstrate a statistically significant improvement in median PFS with nivolumab plus SOC versus SOC (11.9 months vs 11.9 months [HR, 0.81, 95% CI, 0.53 to 1.23; $p=0.30$], respectively). However, the backbone chemotherapy was FOLFOX and 57% of patients in the nivolumab plus SOC arm were RAS wild-type. Discouraging results also come from a phase Ib single-arm clinical trial (Herting CJ et al. *Cancer Immunol Immunother* 2021) in which, in a first-line setting, pembrolizumab plus mFOLFOX6 showed the median PFS of 8.8 months (80% CI, 7.7–11.3 months). These considerations have been added to the discussion.

3. What is the rate for the overall treatment-related (Tr) AEs and also for Grade 3-5-specific? Was the rate of treatment-related AEs higher than non-treatment-related AEs? Also what were the AE rates (Tr AE and irAE) for the two subgroups (pMMR/MSS and dMMR/MSI)?

We thank the reviewer for the comment and the opportunity to clarify.

A total of 449 treatment-related adverse events (TrAEs) of any grade were observed across the three study drugs. Sixty-three out of 73 patients (86.3%) experienced at least one TrAE of any grade, and 48 patients (65.8%) experienced at least one grade 3–5 TrAE. Non-treatment-related AEs occurred in 48 of 73 patients (65.8%), indicating that TrAEs were more frequent than non-treatment-related ones.

Among patients with pMMR/MSS tumors, 57 out of 63 (90.5%) experienced TrAEs of any grade. Among patients with dMMR/MSI tumors, the rate was 100% (10 out of 10). Considering irAEs, 46 of 63 (73.0%) MSS patients and 7 of 10 (70.0%) MSI patients experienced at least one such event.

This information has been added in the updated version of the manuscript in a dedicated paragraph on safety.

4. Questionable translational analysis using Q1 and Q3 PFS. It is unclear how the 25th percentile (Q1) (poor prognosis subgroup) or in 75th percentile (Q3) (good prognosis subgroup) of the PFS values were defined given some patients with censored status. For example, if a patient had a PFS censored in the 1st month, the patient may have a very long PFS if the patient is able to be followed up. In other words, grouping based on PFS value could be biased without considering censored status.

We thank the reviewer for this important suggestion, which allowed us to improve our paper. Based on this request, we updated the PFS status of all 73 patients at 24 months, the end of the study's observation period. Therefore, we no longer have censored PFS data, which certainly strengthened the evidence described.

Another major question is a classification based on “resistance signature” (RES), which was composed of 53 genes altered only in the 25th percentile subgroup. If I interpret it correctly, patients with RES would be only in PFS Q1 group. Accordingly, the survival plot was just a similar comparison between Q1 vs Q2-Q3, not due to RES. Moreover, when patients in the Q1 group were excluded from the analysis, it means the remaining patients should be RES wild-type. How was the analysis conducted in comparison with the mutant vs wildtype. The same concern applies to the Q3 PFS group for the sensitivity signature (SENS). If these issues are not addressed, the remaining translational analysis could be problematic.

We thank the Reviewer for giving us the opportunity to better explain the survival plots accordingly with the RES and SENS signatures. We want to highlight that the PFS quartiles were recalculated in this revised version of the paper, based on the updated PFS data. This led to slight differences

in the distribution of patients among the different quartiles and, therefore, to slight differences in the genes of the two signatures. Nevertheless, the pathways associated with these signatures were similar.

To establish the RES signature, we selected a group of 51 genes altered in the subgroup of patients with poor prognosis (12 patients with PFS below the 25th percentile - Q1 PFS) and not altered in the subgroup of patients with good prognosis (12 patients with PFS over the 75th percentile - Q4 PFS). In this manner, we identified a “RES signature” potentially associated with a worse outcome of MSS patients. Genomic alterations in any of the genes of the RES signature were found also in 16 patients in Q2-Q3 PFS (**Supplementary Figure 3b new**). We next compared the PFS curves of patients with genomic alterations (GAs) in at least one of the 51 genes of the RES signature (RES+) and patients without GAs in any of them (RES-). The 28 RES+ mut patients showed a significantly shorter PFS compared to the 21 RES- patients, with mPFS values of 7.8 months and not reached, respectively ($p < 0.0001$) (**Figure 2b new**).

When we excluded the poor prognosis subgroup (Q1 PFS) from the survival analysis, the remaining 16 RES+ patients in Q2-Q3 (RES+ without the 25th percentile; RES+w/t 25th) showed a worse PFS compared to the 21 RES- patients (mPFS values of 9.6 months and not reached, respectively; $p = 0.0002$) (**Figure 2b new**).

As regards the SENS signature, we identified a “SENS signature” possibly associated to a better outcome of MSS patients, selecting a group of 102 genes altered in the good prognosis subgroup (Q4 PFS) and not altered in the poor prognosis subgroup (Q1 PFS). Genomic alterations in any of the genes of the SENS signature were carried also by patients in Q2-Q3 PFS (**Supplementary Figure 3b new**). We then compared the PFS curves of patients with GAs in at least one of the 102 genes of the SENS signature (SENS+) and patients without GAs in any of them (SENS-). The 30 SENS+ patients showed a significantly better PFS compared to the 19 SENS- patients, with mPFS values of 11.4 and 6.3 months, respectively ($p = 0.0007$). (**Figure 2c new**).

When we excluded the good prognosis subgroup (Q4 PFS) from the survival analysis, the SENS signature showed a trend in separating the PFS curves of SENS+ without the 75th percentile (SENS+w/t 75th) and SENS- patients (mPFS values of 9.30 and 6.30 months, respectively; $p = 0.560$) (**Figure 2c new**).

To further clarify the distribution of patients with at least one mutated gene in the RES and SENS signatures across the quartiles, we added a figure in the Supplementary Information (**Supplementary Figure 3b**).

5. In Figure 5, it is unclear why there are only 19 patients presented to highlight the difference between the 75th vs 25th PFS comparison. The author indicated that 48 samples were profiled with RNA sequencing – so the number should be around 24.

We apologize with the Reviewer for the lack of clarity. We performed RNA-seq analysis on forty-eight samples (42 MSS and 6 MSI) and, as for CGP analysis, we focused our attention only on MSS patients. Among the MSS patients with RNAseq data available, we classified 11 patients in the subgroup with the worse prognosis (25th percentile PFS) and 9 patients in the better prognosis subgroup (75th percentile PFS) based on the updated PFS values. We better clarified our findings in the main text, in the new paragraph Transcriptomic analysis, and improved the legend of Figure 5 (now **Figure 3**) to clarify this point.

6. The need for a comparison with published signatures. Since the RES and SENS appear to be a major point in the study, I would recommend presenting a supplementary table of these genes. The gene symbols are hard to see in supplementary figure S6. A comparison of the sensitivity signature and resistance signature with published signatures associated with primary and metastatic CRC.

We agree with the Reviewer and provided tables with genes belonging to the updated RES and SENS signatures in Supplementary Material (**Supplementary Table 3** and **Supplementary Table 5**, respectively). An image with higher resolution has been replaced in Figure S6 (now **Supplementary Figure 4**), as suggested.

To our knowledge, specific signatures for KRAS/BRAF mutant MSS patients treated with polychemotherapy plus ICIs have not been reported yet. In order to accommodate the Reviewer request, we applied the consensus molecular subtype (CMS) classification and the Intrinsic CMS (iCMS) signature to the cohort of 48 patients (42 MSS and 6 MSI) with available RNAseq data.

The CMS classification is the most validated signature based on gene expression profiling for CRC. However, we want to highlight that this classification is based on the differential gene expression of cohorts of tumors including both RAS/BRAF mutant and wild type CRC, whereas our cohort includes only patients that carried mutations in RAS or BRAF genes.

We obtained the following distribution among CMS classes (**Supplementary Figure 6a**):

	MSI	MSS	Total
CMS1	3	3	6
CMS2	0	14	14
CMS3	0	6	6
CMS4	1	14	15
Not classified	2	5	7

The heatmap of gene expression profile is shown in **Supplementary Figure 6a**:

We also report in **Supplementary Figure 6b** the relative expression of signaling pathways associated with the CMS classification, which is in agreement with previous findings:

Although the total number of cases is limited (n. 41 with known CMS status, 37 MSS and 4 MSI), we found a correlation between the CMS classification and PFS. In particular, patients within the CMS1 group showed a longer mPFS as compared with the other CMS classes (**Supplementary Figure 6c**)

	Time (Months)					mPFS (mos)	Log-rank test
	0	6	12	18	24		
CMS1	6	5	4	4	4	NR	Reference
CMS2	14	13	5	2	2	9.6	p=0.086
CMS3	6	5	1	0	0	8.95	p=0.054
CMS4	15	13	8	5	4	12.9	p=0.199

However, in the cohort of 37 MSS patients with available CMS classification, no significant difference was found among the different CMS classes of CRC (**Supplementary Figure 6d**):

	Number at risk					mPFS (mos)	Log-rank test
CMS1	3	2	1	1	1	8	Reference
CMS2	14	13	5	2	2	9.6	p=0.918
CMS3	6	5	1	0	0	8.95	p=0.640
CMS4	14	12	7	5	4	11.3	p=0.805

We next combined the DNA and RNA data. Sequencing of DNA was successful for 36/37 patients with CMS classification. As shown in **Figure 4a**, detection of genomic alterations belonging to the RES signature identified subgroups with shorter PFS among the CMS2 ($p=0.022$) and CMS4 ($p=0.049$) and, at lesser extent, CMS1 ($p=0.225$) and CMS3 ($p=0.327$) cohorts. Although the numbers of cases are extremely low and prevent any firm conclusion, these preliminary findings suggest that the RES signature can better stratify patients of the NIVACOR cohort as compared with the CMS classification.

	Number at risk					mPFS (mos)	Log-rank test
CMS1 RES +	2	1	0	0	0	6.75	Reference
CMS1 RES -	1	1	1	1	1	NR	p=0.225
CMS2 RES +	8	7	1	0	0	9.25	Reference
CMS2 RES -	6	6	4	2	2	16.2	p=0.022
CMS3 RES +	4	3	0	0	0	7.75	Reference
CMS3 RES -	2	2	1	0	0	10.35	p=0.327
CMS4 RES +	5	4	2	0	0	8.8	Reference
CMS4 RES -	8	8	5	5	4	21.05	p=0.049

We performed a similar analyses for the SENS signature. The PFS of the SENS+ cases was generally longer as compared with the SENS– cases among the different CMS classes, although the difference is higher in the CMS1 group as compared with the other CMS classes (**Figure 4b**).

	Number at risk				mPFS (mos)	Log-rank test
CMS1 SENS+	2	2	1	1	16	Reference
CMS1 SENS-	1	0	0	0	5.5	p=0.157
CMS2 SENS+	9	9	3	1	9.7	Reference
CMS2 SENS-	5	4	2	1	7.9	p=0.842
CMS3 SENS+	2	2	0	0	8.95	Reference
CMS3 SENS-	4	3	1	0	8.1	p=0.654
CMS4 SENS+	11	12	6	5	12.9	Reference
CMS4 SENS-	2	1	1	1	11.45	p=0.256

We also applied to our cohort of patients the iCMS signature that better recapitulates tumor heterogeneity at single cell level.

Among the 48 patients with available RNAseq data, we classified 17 patients (16 MSS and 1 MSI) in the iCMS2 group and 17 patients (15 MSS and 2 MSI) in the iCMS3 group. Fourteen patients were unclassified (**Supplementary Figure 7a**).

No significant differences in term of PFS were observed between the iCMS2 and iCMS3 groups in the whole cohort of 34 patients with iCMS classification (31 MSS and 3 MSI) (**Supplementary Figure 7b**).

The difference in PFS was not significant also when the cohort of 31 MSS patients was analyzed (**Supplementary Figure 7c**).

Finally, we combined the iCMS signature with CGP data in the MSS subgroup. Sequencing of DNA was successful for 30/31MSS patients with the iCMS classification. Detection of genomic alterations in the RES signature identified patients with a significantly low PFS in the iCMS3 group, as compared with RES— patients (6.0 months vs 21.05 months respectively, $p=0.0007$) (**Figure 5a**). No difference was observed in PFS of patients stratified for RES and iCMS2 signatures (9.25 months for patients RES+ versus 9.5 months for patients RES— in the iCMS2 cohort, $p=0.235$) (**Figure 5a**).

	Number at risk					mPFS (mos)	Log-rank test
iCMS2 RES+	10	8	1	0	0	9.25	Reference
iCMS2 RES-	6	6	2	1	2	9.5	p=0.235
iCMS3 RES+	6	3	0	0	0	6.0	Reference
iCMS3 RES-	8	8	6	5	4	21.05	p=0.0007

For the SENS signature, the PFS of the SENS+ cases was higher as compared with the SENS— cases (18.1 months versus 6.1 months, respectively, $p=0.068$) in the iCMS3 class (**Figure 5b**). The difference was less marked for patients stratified for the SENS and the iCMS2 signatures (9.6 months for patients SENS+ versus 7.1 months for patients SENS-, $p=0.454$) (**Figure 5b**).

	Number at risk					mPFS (mos)	Log-rank test
iCMS2 SENS+	10	10	2	1	1	9.6	Reference
iCMS2 SENS-	6	4	1	0	1	7.1	p=0.454
iCMS3 SENS+	7	7	4	4	3	18.1	Reference
iCMS3 SENS-	7	4	2	1	1	6.1	p=0.068

Our results suggest that both RES and SENS signatures better stratified patients in the NIVACOR study than the iCMS classification, although this observation should be confirmed in a larger number of patients.

These findings have been added to the Results section in the paragraph Transcriptomic analysis.

7. The need for high-resolution figures. It is difficult to view Fig 1's swimmer plot. Also, the survival plot needs to include number of at risk and event at each time point.

We thank the reviewer for this suggestion. The Swimmer plot in **Figure 1** has been replaced. The number of patients at risk and events at each time point have been added in the figures of survival curves.

8. Lack of subgroup analysis (pMMR/MSS and dMMR/MSI). While the overall results were presented, it is more informative to include analysis results for both pMMR/MSS and dMMR/MSI

subgroups, such as median treatment duration, median follow-up, similar Fig 1 for each subgroup, and others.

We agree with the reviewer that subgroup analyses are more informative. We added the subgroup comparison into the main text, including the pMMR/MSS, the dMMR/MSI and the RAS/BRAF subgroups analysis (**Supplementary Figure 1 and 2 new**).

9. Since this is a clinical trial report, baseline demographic and disease characteristics should be in the primary table, not in the supplementary table.

We thank the reviewer for raising this relevant issue. In order to accommodate this point, we moved the supplementary table 1 in the main text (**Table 1** in the updated version of the manuscript).

10. The study indicates the population includes RAS/BRAF mutated mCRC in 73 patients. However, from supplementary figure 3, there are ~50 KRAS and ~10 BRAF mutations, which don't add up.

We apologize with the Reviewer for the lack of clarity. We reported in **Supplementary Figure 3a new** only genes with at least 7 genomic alterations identified in 55 patients for which GCP was successfully performed. Regarding KRAS/BRAF mutations, we found 51 KRAS and 7 BRAF mutations (n=58) in 55 patients, including 3 patients with concurrent KRAS and BRAF mutations. We specified in the legend of **Supplementary Figure 3a new** the number of patients analysed by CGP.

11. The bioinformatics method that was applied for the differential gene expression analysis should be more clearly stated, such as if the author applied DEseq2.

We apologize with the Reviewer for the lack of clarity. For differential gene expression analysis, we performed a two group comparison statistics analysis (t-test) using the Qluore omics explorer data analysis software v.3.9. We added this information in the RNA Sequencing data Processing and Analysis section of Methods.

12. Based on the presented analysis, the user should refrain from making inferences related to lymphocyte-T CD8+ and regulation of T-reg cells without direct measurements related to these cells.

We agree with the Reviewer that is difficult to make inferences about lymphocyte-T CD8+ and regulation of T-reg cells without direct measurements. For this reason, we deleted "lymphocyte-T CD8+ and regulation of T-reg cells" from the text.

Reviewer #5 (Remarks to the Author):

We want to thank the Reviewers n. 2 and 3 for their kind responses. We will try to accommodate the concerns of the Reviewer n. 4 as accurately as possible.

Reviewer #4 (Remarks to the Author):

The authors have addressed most of our concerns, including providing a detailed literature review and safety analysis. However, concerns remain regarding the RES and SENS signatures due to potential selection bias:

- **RES Signature:** “To establish the RES signature, we selected a group of 51 genes altered in the subgroup of patients with poor prognosis (12 patients with PFS below the 25th percentile – Q1 PFS) and not altered in the subgroup of patients with good prognosis (12 patients with PFS above the 75th percentile – Q4 PFS).”
- **SENS Signature:** “We selected a group of 102 genes altered in the good prognosis subgroup (Q4 PFS) and not altered in the poor prognosis subgroup (Q1 PFS).”

Both signatures were derived by comparing Q1 (n=12) and Q4 (n=12) PFS groups, leaving approximately 44 patients untested. A more unbiased approach would be to statistically evaluate these remaining patients (n≈44) based on RES and SENS classifications. Figures 2B and 2C may reflect biased results, as they likely include most Q1 patients in the RES+ group and Q4 patients in the SENS+ group, potentially inflating statistical significance. We recommend redoing the survival analysis by excluding Q1 and Q4 groups to reduce bias.

We first want to thank the Reviewer’s for his valuable comments that offered us the opportunity for reducing potential selection bias of the analyses and better explaining our workflow.

We apologize with the Reviewer for the lack of clarity about the number of patients included in the CGP analysis. CGP was successfully performed in 55 patients, including 49 MSS patients. Twelve patients with PFS values within the 25th percentile were classified in the Q1 subgroup and 12 patients with a PFS within the 75th percentile were classified in the Q4 subgroup. The 25 (not 44) remaining patients with PFS values between the 25th and 75th percentiles belonged to the Q2 (n=13 patients) and Q3 (n=12 patients) subgroups.

As shown in figure 2b in the revised version, 21 patients were negative for the RES signature (RES-) and 28 were positive, including the 12 patients in Q1/25th percentile. In order to reduce bias due to the presence of patients with poor outcome, we excluded all patients of the Q1 group from the RES+ population (indicated as RES+ w/o 25th percentile; n=16 patients, including 8 Q2 and 8 Q3 patients). The findings shown in Figure 2b, green curve, suggest that even excluding the poor prognosis population (all Q1 patients) the difference between RES+ and RES- patients is still statistically significant.

Similarly, we excluded patients with good prognosis (Q4, 75th percentile) from the SENS+ group (Figure 2c, green line). The PFS of the SENS+ w/o 75th percentile group (n=19 patients, including 12 Q2 and 7 Q3 patients) was not statistically different when compared with the PFS of the SENS- population, although a trend was observed.

We defined more clearly the distribution of patients in the Q2-Q3 PFS subgroups in the text in order to address the reviewer’s concern.

A similar concern applies to the stratification analysis of RES and SENS within CMS and iCMS subtypes. If Q1 patients disproportionately contribute to the RES+ group in CMS2 and iCMS3, this could also lead to biased statistical significance.

Regarding the correlations of the RES and SENS signatures with the CMS and iCMS classification, we agree with the Reviewer that patients in Q1 and Q4 might lead to biased statistical significance. To accommodate this point, we performed the analysis excluding patients in the Q1 subgroup from the survival analysis of the RES signature within CMS and iCMS subtypes, and the Q4 subgroup from the analysis of SENS signature within CMS and iCMS subtypes, although the number of patients classified in CMS/iCMS subgroups was very low and might lack statistical significance.

The combined analysis of RES and CMSs signatures after excluding patients in the Q1/25th percentile showed results similar to the analyses of the whole population. Although the difference in PFS between RES+ and RES- was not statistically significant, RES- patients showed longer PFS as compared with RES+ patients in the CMS2 and CMS4 classes (Supplementary Fig. 7a in the new Supplementary Information file).

	Number at risk				mPFS (mos)	Log-rank test
CMS1 RES+	1	1	0	0	8	Reference
CMS1 RES-	1	1	1	1	NR	p=0.317
CMS2 RES+	6	6	1	0	9.6	Reference
CMS2 RES-	6	6	4	2	16.2	p=0.055
CMS3 RES+	2	2	0	0	9.75	Reference
CMS3 RES-	2	2	1	0	10.35	p=0.695
CMS4 RES+	4	4	2	0	10.85	Reference
CMS4 RES-	8	8	5	5	21.05	p=0.108

Similarly, the combined analysis of SENS and CMS signatures revealed no statistical significant differences when patients belonging to the Q4 subgroup were excluded from the analysis. However, the results showed a trend similar to those obtained when including the whole population of patients (Supplementary Fig. 7b in the new Supplementary Information file).

	Number at risk				mPFS (mos)	Log-rank test	
CMS1 SENS+	1	1	0	0	0	8	Reference
CMS1 SENS-	1	0	0	0	0	5.5	p=0.317
CMS2 SENS+	8	8	2	0	0	9.6	Reference
CMS2 SENS-	4	3	1	0	0	7.1	p=0.354
CMS3 SENS+	2	2	0	0	0	8.95	Reference
CMS3 SENS-	4	3	1	0	0	8.1	p=0.654
CMS4 SENS+	6	6	1	0	0	8.25	Reference
CMS4 SENS-	2	1	1	0	0	11.45	p=0.457

We also excluded the Q1 subgroup from the RES and iCMS signature combined analysis. No statistically significant differences were observed between the iCMS2/3 RES+ and iCMS2/3 RES- patients after removing patient in Q1. However, the PFS of the iCMS3 RES- patients was much longer as compared with the iCMS RES+ patients, in agreements with the results of the whole population (Supplementary Fig. 9a in the new Supplementary Information file).

	Number at risk				mPFS (mos)	Log-rank test	
iCMS2 RES+	7	7	1	0	0	9.7	Reference
iCMS2 RES-	6	6	2	1	1	9.5	p=0.454
iCMS3 RES+	1	1	0	0	0	9.4	Reference
iCMS3 RES-	8	8	6	5	4	21.05	p=0.068

Similar results were observed when we excluded the Q4 subgroup from the SENS and iCMS signature analysis. The PFS of the SENS+ patients was slightly longer when compared with the SENS- cases within both iCMS classes (Supplementary Fig. 9b in the new Supplementary Information file).

	Number at risk				mPFS (mos)	Log-rank test	
iCMS2 SENS+	9	9	1	0	0	9.5	Reference
iCMS2 SENS-	6	4	1	0	0	7.1	p=0.977
iCMS3 SENS+	3	3	0	0	0	8.5	Reference
iCMS3 SENS-	6	3	1	0	0	6	p=0.3965

These figures have been added to the supplementary information file (Supplementary Figures 7a-b and 9a-b) and the text was accordingly updated.

We hope that these additional analyses have addressed the reviewer's concerns.